# Efficient Top-$m$ Data Values Identification for Data Selection

**Xiaoqiang Lin[1][*], Xinyi Xu[1], See-Kiong Ng[2], Bryan Kian Hsiang Low[1]**
[1]Department of Computer Science, National University of Singapore, Singapore
[2]Institute of Data Science, National University of Singapore, Singapore
`xiaoqiang.lin@u.nus.edu`, `xinyi.xu@u.nus.edu`,
`seekiong@nus.edu.sg`, `lowkh@comp.nus.edu.sg`

## Abstract

Data valuation has found many real-world applications, e.g., data pricing and data selection. However, the most adopted approach – Shapley value (SV) – is computationally expensive due to the large number of model trainings required. Fortunately, most applications (e.g., data selection) require only knowing the $m$ data points with the highest data values (i.e., top-$m$ data values), which implies the potential for fewer model trainings as exact data values are not required. Existing work formulates top-$m$ Shapley value identification as top-$m$ arms identification in multi-armed bandits (MAB). However, the proposed approach falls short because it does not utilize data features to predict data values, a method that has been shown empirically to be effective. A recent top-$m$ arms identification work does consider the use of arm features while assuming a linear relationship between arm features and rewards, which is often not satisfied in data valuation. To this end, we propose the GPGapE algorithm that uses the Gaussian process to model the *non-linear* mapping from data features to data values, removing the linear assumption. We theoretically analyze the correctness and stopping iteration of GPGapE in finding an $(\varepsilon, \delta)$-approximation to the top-$m$ data values. We further improve the computational efficiency, by calculating data values using small data subsets to reduce the computation cost of model training. We empirically demonstrate that GPGapE outperforms other baselines in top-$m$ data values identification, noisy data detection, and data subset selection on real-world datasets. We also demonstrate the efficiency of our GPGapE in data selection for large language model fine-tuning.

## 1 Introduction

Data is essential to obtaining a good-performing machine learning (ML) model. Data valuation (Ghorbani & Zou, 2019) quantifies the contribution of each data point to the model performance. The contribution estimate (i.e., data value) can be used in data pricing (Agarwal et al., 2019), data debugging by identifying noisy data (Koh & Liang, 2017; Kwon & Zou, 2021), data selection (Nohyun et al., 2022), and incentive design in collaborative machine learning (CML) (Sim et al., 2020; Xu et al., 2021). The most commonly adopted data valuation approaches (Ghorbani & Zou, 2019; Jia et al., 2019b) are based on the Shapley value (SV) due to its desirable properties (e.g., fairness) and good empirical performance in data selection (Ghorbani & Zou, 2019; Wang et al., 2024). Specifically, the data value, precisely SV, is defined as the (average of) changes in model performance (i.e., marginal contribution) when the data point is removed from different subsets of the training dataset (see Equ. (1)). However, the computation of exact data values requires $n!$ (where $n$ is # data points) model trainings, posing a significant challenge to applying it to real-world large datasets. Although existing approaches have explored several sampling-based approximations (Ghorbani & Zou, 2019; Okhrati & Lipani, 2021), the computational cost remains high, especially when complex models such as neural networks (NNs) are used.

Fortunately, most applications only require knowing the $m$ data points with the highest data values (i.e., top-$m$ data values). For example, in data marketplaces, buyers with limited budgets will only

---

[*]Corresponding authors.

buy data with the largest $m$ data values (Ghorbani et al., 2022). In noisy data detection, noisy data are specified by data with $m$ lowest data values (Wang et al., 2020; Schoch et al., 2022). In CML, some incentive designs reward only the top-$m$ highest contributing participants (Zhang et al., 2021). Certain fairness properties are preserved when using top-$m$ data values in incentive design (see Appendix A.5). Intuitively, identifying the top-$m$ data values can incur a lower computational cost than directly approximating data values since *it does not require either approximating the exact data values well or knowing the exact ranking of these data values.* **How to obtain top-$m$ data values efficiently without directly approximating the exact data values?**

An existing work (Ghorbani et al., 2022) has empirically shown that *data features are predictive of the data value of the corresponding data point*. Intuitively, data points with similar data features will have similar data values (see Lemma A.1). On the other hand, existing work (Kolpaczki et al., 2021) has proposed to use the top-$m$ arms identification in multi-armed bandits (MAB) to identify top-$m$ players (via Shapley value) in cooperative games. However, the proposed algorithm does not use the data features and hence fails to identify top-$m$ data values efficiently (see Sec. 5). The work of Réda et al. (2021) proposes top-$m$ arms identification using linear bandit that assumes a linear relationship between arm features (i.e., data features here) and rewards (i.e., data values). *This assumption does not apply to highly complex functions such as the function mapping data features to data values*, especially when the datasets are highly complex (e.g., image datasets). Therefore, its theoretical results and empirical efficiency are not applicable to data valuation (as empirically demonstrated in Sec. 5.1).

To this end, building on Réda et al. (2021), we propose our GPGapE algorithm to identify top-$m$ data values, which uses the *Gaussian process (GP) (Seeger, 2004) to model highly complex and non-linear functions (Seeger, 2004; Bui et al., 2016)*, i.e., the mapping of data features to data values. We theoretically analyze the correctness of GPGapE in identifying an $(\varepsilon, \delta)$-approximation to the top-$m$ data values and provide a worst-case upper bound on the stopping iteration (i.e., $\mathcal{O}(n \log^{d+1} n)$ where $d$ is the dimension of the GP input).

On the other hand, the diminishing return of data in ML models has been observed in real-world datasets (Beleites et al., 2013; Mahajan et al., 2018). To elaborate, the improvement of model performance (i.e., marginal contribution) is less when adding a data point to a large dataset compared to a small one (see Fig. 1). The exact data values require computing the marginal contributions to data subsets of all sizes (including the large ones). Therefore, *repeated model trainings on large data subsets are performed*, but *contribute little to the final data values* since the magnitudes of these marginal contributions tend to be small. Existing work (Ghorbani & Zou, 2019) accelerates data value approximation by discarding the marginal contributions to large data subsets. However, *it is unclear how to utilize this observation to accelerate the top-$m$ data values identification.* We propose to define the data values on small subsets and draw a connection between the top-$m$ data values on small subsets identification and the top-$m$ data values identification. Empirical results show that this approach improved GPGapE by $16.5\times$ in query efficiency w.r.t. marginal contributions (see Table 1) and $1.91\times$ in running time (see Table 2). Overall, GPGapE achieves up to $50\times$ better in query efficiency (see Table 1) compared to existing approaches in achieving the same quality of top-$m$ data values identification. Our contributions are:

- Proposing the GPGapE algorithm that uses the *Gaussian process* to effectively model the function mapping data features to data values to identify top-$m$ data values.
- Analyzing the *correctness* of GPGapE in getting an $(\varepsilon, \delta)$-approximation and establishing a *near-linear* upper bound of stopping iteration.
- Defining data values on small subsets and drawing its connection to top-$m$ identification of the original data values which is used to further accelerate our GPGapE.
- Empirically showing that GPGapE outperforms other data value approximations in top-$m$ data values identification, noisy data detection, and data subset selection.

## 2 SETTING AND PRELIMINARIES

### 2.1 DATA VALUATION AND SHAPLEY VALUE

The most adopted definitions of data values use SV (Ghorbani & Zou, 2019) and its variants (e.g., Banzhaf value (Wang & Jia, 2023) and other semivalues (Kwon & Zou, 2021)) since they provide

desirable properties (e.g., symmetry, strict desirability (Ghorbani & Zou, 2019; Sim et al., 2020)). Denote an index set as $N := \{1, 2, \ldots, n\}$ and a dataset $D_N := \{z_i\}_{i \in N}$ where $z_i := (x_i, y_i)$ is a data point with $x_i \in \mathcal{X}, y_i \in \mathcal{Y}$. SV for a data point $z_i$ is:

$$\varphi_i := \sum_{l=1}^{n} \frac{w(l)}{n} \sum_{\substack{S \subseteq N \setminus \{i\} \\ |S| = l-1}} [U(S \cup \{i\}) - U(S)] \tag{1}$$

where $w(l) = 1/\binom{n-1}{l-1}$ and $U : 2^N \mapsto \mathbb{R}$ is a utility function. Specifically, $U(S)$ measures the utility of the data subset $D_S, S \subseteq N$ and is usually defined as the validation performance of the model trained on data subset $D_S$ (Ghorbani & Zou, 2019). Put simply, SV of $z_i$ is a weighted average of its marginal contributions (i.e., $U(S \cup \{i\}) - U(S)$) to different data subsets $D_S$.

**Probabilistic formulation of SV.** SV defined in Equ. (1) can be rewritten as:

$$\varphi_i = \mathbb{E}_{S \sim P_w}[U(S \cup \{i\}) - U(S)] \tag{2}$$

where $P_w$ is a discrete distribution over $S \subseteq N \setminus \{i\}$ with the probability of $S$ being sampled as $\frac{w(|S|+1)}{n}$. Therefore existing works apply Monte-Carlo (Maleki et al., 2013; Ghorbani & Zou, 2019) to approximate SV. The semivalue is defined as any $w(l)$ that satisfies $\sum_{l=1}^{n} \binom{n-1}{l-1} w(l) = n$. Some works define data values using other semivalues (Kwon & Zou, 2021; Wang & Jia, 2023), but we restrict our discussion to SV and thus use SV and data value interchangeably for simplicity. Our approach is applicable to all semivalues with minor changes.[1]

## 2.2 SETTINGS

We assume that the data value of $z_i$ is labeled by a function $f : \mathcal{X} \times \mathcal{Y} \to \mathbb{R}$ mapping data points to data values, i.e., $f(z_i) = \varphi_i$. For notational simplicity and w.l.o.g., we arrange data points such that $\varphi_1 \geq \varphi_2 \geq \cdots \geq \varphi_m > \varphi_{m+1} \geq \varphi_{m+2} \geq \cdots \geq \varphi_n$. We assume that $\varphi_m > \varphi_{m+1}$ to guarantee the uniqueness of the exact top-$m$ data values. Since data value is the expectation of marginal contributions (i.e., $U(S \cup \{i\}) - U(S)$), we view a marginal contribution as a noisy observation of the data value $\varphi_i$. Specifically, at each time step $t$ we select a data point $z_t := z_i$ to query its marginal contribution: $U(S \cup \{i\}) - U(S) = f(z_i) + \eta_t$ where the randomness of noise $\eta_t$ comes from the random sampling of the subset $S$ from $P_w$ at time $t$. Denote $\mathcal{S}_m^{*,\varepsilon} := \{a \in N : \varphi_a \geq \varphi_m - \varepsilon\}$ which contains no less than $m$ elements. Denote $\mathcal{S}_m^* := \mathcal{S}_m^{*,0}$ as the top-$m$ data values. Denote the output of an algorithm as $\hat{\mathcal{S}}_m$.

**Definition 2.1** (($\varepsilon, \delta$)-approximation to the top-$m$ data values). [2] An algorithm gives an ($\varepsilon, \delta$)-approximation to top-$m$ data values if its output $\hat{\mathcal{S}}_m$ satisfies $\hat{\mathcal{S}}_m \subseteq N, |\hat{\mathcal{S}}_m| = m$, and $\mathbb{P}(\hat{\mathcal{S}}_m \subseteq \mathcal{S}_m^{*,\varepsilon}) \geq 1 - \delta$.

Our objective is to obtain an ($\varepsilon, \delta$)-approximation to the top-$m$ data values with as few marginal contributions (i.e., queries) as possible to reduce the computational cost of model training.

**Gaussian process (GP)** Let $k(\cdot, \cdot)$ be a kernel function. Assume the initial prior distribution for a function $f$ over the dataset $D \subset \mathbb{R}^d$ for GP is $\mathcal{GP}_D(0, v^2 k(\cdot, \cdot))$ where $v$ is a scaling parameter. Given query points $(z_1, z_2, \ldots, z_t)$ in domain $D$ with observations $y'_{1:t} = [y'_1, \ldots, y'_t]^T$, vector $k_t(z) = [k(z_1, z), \ldots, k(z_t, z)]^T$, matrix $K_t = [k(z_i, z_j)]_{i,j=1}^t$, and noise parameter $\lambda$ for GP, the posterior over $g$ at iteration $t$ is $\mathcal{GP}_D(\mu_t(\cdot), v^2 k_t(\cdot, \cdot))$, where

$$\mu_t(z) := k_t(z)^T (K_t + \lambda I)^{-1} y'_{1:t}, \quad k_t(z_i, z_j) := k(z_i, z_j) - k_t(z_i)^T (K_t + \lambda I)^{-1} k_t(z_j) . \tag{3}$$

The GP model is shown to be able to model complex functions with different selections of the kernel function $k(\cdot, \cdot)$ and is a key component in our GPGapE algorithm.

**$R$-sub-Gaussian** The distribution of a random variable $X$ is $R$-sub-Gaussian if $\mathbb{E}[e^{\alpha X}] \leq \exp((\alpha^2 R^2)/2), \forall \alpha \in \mathbb{R}$.

---

[1] This is achieved by adjusting the sampling probability to $\frac{w'(|S|+1)}{n}$, where $w'(\cdot)$ specifies the semivalue.

[2] The randomness can be from the algorithm or the random sampling of marginal contributions.

# 3 RELATED WORKS

**SV and data value approximation.** Existing works have proposed several approaches to directly approximate SV. For example, the works of Maleki et al. (2013); Ghorbani & Zou (2019) propose a Monte-Carlo-based SV approximation. Other similar sampling-based approaches are proposed (Castro et al., 2017; Okhrati & Lipani, 2021; Mitchell et al., 2022). The work of Covert & Lee (2021) proposes a regression-based SV approximation and Li & Yu (2023) further improves upon it. The work of Kolpaczki et al. (2024) proposes an SV approximation without dependency on marginal contributions. Some other works approximate data values using specific characteristics of ML. Specifically, the works of Jia et al. (2019a); Wang et al. (2023); Castro et al. (2017) propose model-specific approximations to data values. However, they are only applicable to the $k$-nearest neighbor model (or its variants). The work of Jia et al. (2019b) proposes a group testing-based data value approximation. However, its theoretical result is w.r.t. the $l2$ norm approximation to SV under bounded utility assumption which is not applicable here. In general, these works approximate data values directly and hence their efficiency in identifying top-$m$ data values is unclear, which we will show in Sec. 5. Some other data value approximations rely on utility approximation (Wang et al., 2021; Wu et al., 2022), and hence are complementary with our work since they can be used to further accelerate our GPGapE.

**Top-$m$ SVs identification and data selection.** The work of Suri & Narahari (2008) proposes to identify the top-$m$ nodes in social network via identifying top-$m$ SVs. However, they use Monte-Carlo approximation which is not effective. The work of Kolpaczki et al. (2021) first proposes to identify top-$m$ SVs using MAB. However, it is not for identifying top-$m$ data values and hence does not use the data features, failing to identify the top-$m$ data values efficiently (see Sec. 5). Our work is the first to study the problem of top-$m$ data values identification and propose the GPGapE algorithm that uses data features with theoretical analysis. The work of Lin et al. (2023; 2024); Wang et al. (2024) study the use of Shapley value or other semivalues to identify helpful data points in the dataset and select a data subset of size $m$ to achieve better or comparable performance to the full dataset. Other works (Chen et al., 2024; 2025) have attempted to find the best data subset using Bayesian optimization and bandit algorithms.

**Multi-armed bandits (MAB).** The majority of MAB works consider the best arm identification (Camilleri et al., 2021; Zhu et al., 2021) instead of top-$m$ arms identification. The work of Kalyanakrishnan & Stone (2010); Kalyanakrishnan et al. (2012) explore the extension of best arms identification to top-$m$ arms identification and recent work (Réda et al., 2021) has improved the query efficiency by considering top-$m$ arms identification in linear bandit. However, the work of Réda et al. (2021) requires the linear assumption between arm features and rewards, hence making its theoretical results and empirical efficiency not applicable to data valuation (see Sec. 5). The work of Mason et al. (2022) studies the level set estimation problem which aims to find arms with rewards more than a specific value, hence a different problem from our top-$m$ data values identification. Our work extends the work of Réda et al. (2021) by using GP to model the underlying mapping function to better utilize the data features for top-$m$ data values identification and obtain new theoretical results.

# 4 TOP-$m$ DATA VALUES IDENTIFICATION

We will describe our GPGapE algorithm and its theoretically analysis in Sec. 4.1. After that, we will discuss how to further accelerate the top-$m$ data values identification by defining the data values on small subsets and drawing the connection between identifying top-$m$ data values on small subsets and top-$m$ data values in Sec. 4.2.

## 4.1 GPGAPE ALGORITHM

We introduce the GPGapE algorithm, an adaptation of the top-$m$ linear bandit algorithm (i.e., $m$-LinGapE (Réda et al., 2021)) to use the Gaussian process (GP) to model the function mapping data features to data values, replacing the original linear model.

Our GPGapE requires the *gap index*, which is an upper confidence bound on the difference of data values between two data points. Specifically, denote the true gap of the data values between two data

points $z_i$ and $z_j$ as $G(z_i, z_j) := \varphi_i - \varphi_j$ and its estimates at time $t$ as $\hat{G}_t(z_i, z_j) := \mu_t(z_i) - \mu_t(z_j)$. Denote $\sigma_t^2(z) := k_t(z, z)$. The gap index $B_t(z_i, z_j)$ is defined as:

$$B_t(z_i, z_j) := \hat{G}_t(z_i, z_j) + C_{\delta,t} W_t(z_i, z_j), \quad W_t(z_i, z_j) := \sqrt{\sigma_t^2(z_i) + \sigma_t^2(z_j) - 2k_t(z_i, z_j)} \quad (4)$$

where $C_{\delta,t}$ is a weighting parameter discussed in Theorem 4.1. Intuitively, $\hat{G}_t(z_i, z_j)$ is the estimation of the gap of two data values $\varphi_i$ and $\varphi_j$ using the GP posterior mean in time $t$ and $W_t(z_i, z_j)$ is the standard deviation of the gap estimate. Therefore, the gap index is an upper confidence bound of the gap between $\varphi_i$ and $\varphi_j$. The gap index is crucial in the design of our algorithm when actively finding the next data point to compute its marginal contributions. Define $G_i := \varphi_i - \varphi_{m+1}$ if $i \le m$, $\varphi_m - \varphi_i$ otherwise. Denote $\arg\max_{j \in N}^{[m]} \mu_t(z_j)$ the indices in $N$ with top-$m$ $\mu_t(z_j)$. Denote the stopping iteration for GPGapE as $\tau_\delta$, the pseudo-code for GPGapE is in Algorithm 1.

---

**Algorithm 1** GPGapE for top-$m$ data values identification

---

**input** $\{z_i\}_{i \in N}$: Data points to be evaluated; $\varepsilon$: Stopping threshold; $m$: Number of largest data values to be identified; $\delta$: Parameter for $C_{\delta,t}$; $U : 2^N \to \mathbb{R}$, utility function; $\lambda$: Noise parameter for GP.
1: $t \leftarrow 0$
2: **repeat**
3:     $t \leftarrow t + 1$
4:     Select candidate set for top-$m$ data values: $J(t) \leftarrow \arg\max_{j \in N}^{[m]} \mu_{t-1}(z_j)$
5:     $b_t = \arg\max_{j \in J(t)} \max_{i \notin J(t)} B_{t-1}(z_i, z_j)$
6:     $c_t = \arg\max_{a \notin J(t)} B_{t-1}(z_a, z_{b_t})$
7:     Decide the data point to query its marginal contribution: $a_t = \arg\max_{i \in \{b_t, c_t\}} \sigma_{t-1}(z_i)$
8:     $y_t' = U(S \cup \{a_t\}) - U(S)$ where $U(S)$ is obtained by training a ML model on a sampled $D_S$
9:     Update GP with data feature-marginal contribution pairs $\{(z_{a_1}, y_1'), ..., (z_{a_t}, y_t')\}$ (see Equ. (3))
10:     $\tau_\delta \leftarrow t$
11: **until** $B_t(z_{c_t}, z_{b_t}) \le \varepsilon$
12: **return** The identified top-$m$ data values: $\hat{S}_m^{\tau_\delta} \leftarrow \arg\max_{j \in N}^{[m]} \mu_t(z_j)$

---

To elaborate, at time $t$, we select the top-$m$ data points with the largest GP posterior mean as the candidate set $J(t)$ (in line 4 of Algorithm 1). After that, we find $b_t \in J(t)$ and $c_t \in N \setminus J(t)$ such that $B_t(z_{c_t}, z_{b_t})$ is maximized. Intuitively, $B_t(z_{c_t}, z_{b_t})$ is the upper confidence bound of $\varphi_{c_t} - \varphi_{b_t}$. A larger $B_t(z_{c_t}, z_{b_t})$ means that $z_{c_t}$ potentially has a high data value but has not been selected in $J(t)$ and hence challenges the potentially low data value data point $z_{b_t}$ from the candidate set $J(t)$. Therefore, more marginal contributions are needed for these two data points (decided by the GP posterior variance in line 7) to get more information for improving candidate set $J(t)$. The subset $S \subseteq N \setminus \{a_t\}$ is sampled from $P_w$ (described in Sec. 2). GP posterior is updated using the data feature-marginal contribution pairs $\{(z_{a_1}, y_1'), \ldots, (z_{a_t}, y_t')\}$ and hence is able to model the function mapping from data points to data values. Our GPGapE stops when the stopping condition holds and outputs $\hat{S}_m^{\tau_\delta}$ as the identified top-$m$ data values. We theoretically show that $\hat{S}_m^{\tau_\delta}$ is an $(\varepsilon, \delta)$-approximation to top-$m$ data values under the stopping condition in Algorithm 1.

**Theorem 4.1** (Correctness of GPGapE). Assume that $\{\eta_t\}_{t=1}^\infty$ are $R$-sub-Gaussian. Let $k(\cdot, \cdot)$ be a positive-semidefinite kernel function and let $\delta \in (0, 1]$. Assume that $f$ is a member of the reproducing kernel Hilbert space (RKHS) corresponding to the kernel function $k$ with RKHS norm bounded by $B$. With probability at least $1 - \delta$, the output of our GPGAPE algorithm (when the stopping condition holds) satisfies $\hat{S}_m^{\tau_\delta} \in S_m^{*,\varepsilon}$ when the parameter $C_{\delta,t} = B + R\sqrt{2(\gamma_t + 1 + \ln(1/\delta))}$ where $\gamma_t$ is the maximum information gain (Srinivas et al., 2010) after $t$ steps and the parameter $\lambda$ in GP is set to be $1 + 2/\tau_\delta$.

The proof is in Appendix C. Our assumption on $\{\eta_t\}_{t=1}^\infty$ is reasonable under data valuation. Specifically, the distribution of the marginal contribution is sub-Gaussian when $U$ is validation accuracy. To elaborate, for a bounded random variable within $[a, b]$, the variable is $\frac{b-a}{2}$-sub-Gaussian (Arinaldo, 2018). In our case, the marginal contribution is the random variable, and the utility function outputs are within $[0, 1]$ when it is validation accuracy. Since the marginal contribution is the difference between two evaluations of the utility function, it will be within $[-1, 1]$. Consequently, the distribution of marginal contribution defined by validation accuracy is trivially

1-sub-Gaussian. Since $\eta_t = U(S \cup \{a_t\}) - U(S) - f(z_{a_t})$ (i.e., marginal contribution shifted by a constant mean), it is also sub-Gaussian when the marginal contribution is sub-Gaussian. Moreover, even if other utility functions (e.g., negative cross-entropy loss for classification) are used, which are not necessarily bounded, our result holds as long as the marginal contribution is sub-Gaussian.

As for the assumption for the mapping function $f$, when $k(\cdot, \cdot)$ is specified as a non-linear kernel function (e.g., radial basis function kernel, Matérn kernel), the function that lives in its corresponding RKHS can be highly non-linear and complex. Moreover, existing works have shown that $f$ in RKHS specified by some special kernels, the outputs of GP resemble NN outputs (Arora et al., 2019). Therefore, our GPGapE is applicable to highly complex functions. As a result, our theoretical result is not restricted to linear functions as in Réda et al. (2021). We provide more detailed discussions and empirical verifications on why GP is a good design choice for modeling the function mapping from data features to data values in Appendix A. Let $\mathbb{R}^+$ denote the set of positive real values.

**Theorem 4.2** (Upper bound of the stopping iteration $\tau_\delta$). Given that the assumptions in Theorem 4.1 hold, with probability at least $1 - \delta$, the stopping iteration $\tau_\delta$ of our GPGapE algorithm satisfies

$$\tau_\delta \leq \inf\{u \in \mathbb{R}^+ : u > 1 + \sum_{a \in N} 12C_{\delta,u}^2 / \max(\varepsilon, \frac{\varepsilon + G_a}{3})^2\} . \tag{5}$$

The proof of Theorem 4.2 is in Appendix C. From Equ. (5), if $G_a$ is large for all $a \in N$, a smaller $u$ is needed for the inequality to hold. Consequently, fewer iterations are needed for the algorithm to learn an $(\varepsilon, \delta)$-approximation to the top-$m$ data values. Intuitively, large $G_a$ means that all other data values are very far away from the $m$-th and $(m + 1)$-th data values, making it easier to identify the top $m$-data values. On the other hand, from Equ. (5), a better approximation (i.e., a smaller $\varepsilon$) requires performing more iterations of the algorithm.

**Proposition 4.3** (Query complexity of GPGapE). Let $D \subset \mathbb{R}^d$ (defined in Sec. 2) be compact and convex. Assume that the kernel function satisfies $\forall z, z', k(z, z') \leq 1$. Given that the assumptions in Theorem 4.1 hold, $\tau_\delta = \mathcal{O}(n \log n)$ if $k$ is the linear kernel function and $\tau_\delta = \mathcal{O}(n \log^{d+1}(n))$ if $k$ is the radial basis function (RBF).

The proof is in Appendix C. Proposition 4.3 gives the query complexity of our GPGapE w.r.t. # data points $n$. Note that since we only evaluate the utility function once in each iteration, the stopping iteration equals the number of total queries to utility functions. It shows that our algorithm is efficient with a near-linear complexity in the worst case ($\mathcal{O}(n \log^{d+1}(n))$ when RBF is used). Note that in Theorem 4.2, the upper bound is problem-dependent. Put differently, it studies how changes in the parameters, $\varepsilon, G_a$, for the problem itself affect the efficiency of our algorithm. While Proposition 4.3 does not focus on problem-dependent parameters (by bounding them with some constants) and gives a result on how the query complexity scales w.r.t. $n$. We empirically verify the efficiency of our GPGapE in Sec. 5.

## 4.2 ACCELERATION BY QUERYING MARGINAL CONTRIBUTIONS ONLY ON SMALL SUBSETS

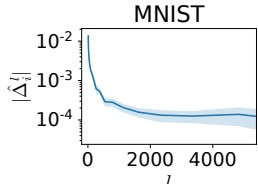 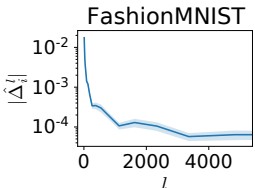 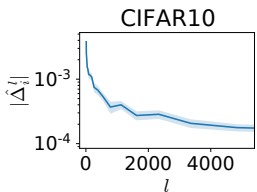

Figure 1: Diminishing return of adding a randomly selected data point $i$ to the data subset when the size of the data subset increases. Marginal contributions are computed via the validation accuracy (details in Appendix A).

ML models are known to have diminishing returns (Beleites et al., 2013; Mahajan et al., 2018), meaning that adding new data to a larger dataset will have a lower benefit (e.g., a lower increase in model accuracy) than adding the same data to a smaller dataset. We exploit this property to further accelerate our GPGapE. Specifically, we denote $\Delta_i^l := \mathbb{E}_{S \subseteq N \setminus \{i\}, |S|=l-1}[U(S \cup \{i\}) - U(S)]$ which is the expected marginal contribution of the data point $z_i$ to $(l - 1)$-sized data subsets. SV

can be rewritten as $\varphi_i = n^{-1} \sum_{l=1}^{n} \Delta_i^l$. In this case, SV is computed as the average of expected marginal contribution $\Delta_i^l$ over different sizes of $S$. Empirically (see Fig. 1), we observe that $|\Delta_i^l|$ is monotonically decreasing w.r.t. $l$ and $\Delta_i^l$ will be close to 0 when $S$ is large. This inspires us to define:

$$\varphi_i(p) := (1/p) \sum_{l=1}^{p} \Delta_i^l \tag{6}$$

where $p \leq n$. $\varphi_i(p)$ only averages the marginal contributions to data subsets with sizes no larger than $p$ (i.e., data value on small subsets).

**Assumption 4.4.** Assume that $\exists p \in N$ such that $|\Delta_i^l| \leq \varepsilon', \forall l \in \{p, \ldots, n\}, \forall i \in N$.

Assumption 4.4 assumes that the return $|\Delta_i^l|$ of a data point $i$ be less than $\varepsilon'$ when the size of the data subset $l$ is larger than $p$. Note that this does not require the return to be monotonically decreasing w.r.t. $l$ but just not exceed a certain value after a certain size. Consequently, this assumption is looser than the diminishing return, making it easier to hold empirically. We draw the following connection given the assumption above holds:

**Proposition 4.5** (Connection between the top-$m$ identification of $\{\varphi_i\}_{i \in N}$ and top-$m$ identification of $\{\varphi_i(p)\}_{i \in N}$)**.** Given that Assumption 4.4 holds, an $(\frac{n}{p}\varepsilon - \frac{2(n-p)}{p}\varepsilon', \delta)$-approximation of top-$m$ of $\{\varphi_i(p)\}_{i \in N}$ is an $(\varepsilon, \delta)$-approximation of top-$m$ of $\{\varphi_i\}_{i \in N}$.

*Remark* 4.6. When $\varepsilon'$ is approaching 0, it seems that the former problem becomes easier than the latter since $\frac{n}{p}\varepsilon \geq \varepsilon$. This is because the magnitude of the $\{\varphi_i(p)\}_{i \in N}$ is larger than the $\{\varphi_i\}_{i \in N}$ in general. On the other hand, when $\varepsilon' \leq \varepsilon/2$, $\frac{n}{p}\varepsilon - \frac{2(n-p)}{p}\varepsilon'$ is monotonically decreasing w.r.t. $p$. This implies that a smaller $p$ will make the problem easier (due to higher $\frac{n}{p}\varepsilon - \frac{2(n-p)}{p}\varepsilon'$) while still maintaining its equivalence to an $(\varepsilon, \delta)$-approximation of $\{\varphi_i\}_{i \in N}$.

The proof for Proposition 4.5 is in Appendix C. From Proposition 4.5, we can run our GPGapE on identifying top-$m$ of $\{\varphi_i(p)\}_{i \in N}$ to obtain an $(\varepsilon, \delta)$-approximation of $\{\varphi_i\}_{i \in N}$, meaning that only marginal contributions for small datasets are required. Specifically, for a data point $z_{a_t}$, a data subset $D_S \subseteq D_N \setminus \{z_{a_t}\}, |D_S| \leq p$ is sampled with probability $\frac{w(|S|+1)}{p}$ in line 7 of Algorithm 1. The training time complexity of the ML model is usually $O(n^2)$ (e.g., kernelized support vector machine) and $O(n^3)$ (e.g., kernel ridge regression). Therefore, the expected time complexities of computing marginal contribution are $O(n^2)$ and $O(n^3)$ respectively. This implies that if $p$ is selected as $\lfloor n/10 \rfloor$, the expected time complexity will potentially be reduced by $100\times$ and $1000\times$ respectively. Surprisingly, as we will see in Sec. 5, with the same number of marginal contribution computations, querying marginal contributions on small subsets will not only reduce the running time but also perform better than the original GPGapE algorithm.

## 5 EXPERIMENTS

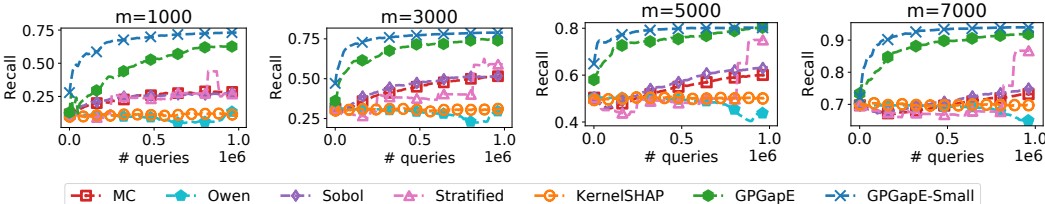

Figure 2: Recall of top-$m$ data values using different approximation approaches.

We demonstrate the effectiveness of GPGapE through experiments on top-$m$ data values identification (Sec. 5.1), noisy data detection (Sec. 5.2) (Wang et al., 2020; Schoch et al., 2022), and data subset selection (Sec. 5.3) (Ghorbani & Zou, 2019; Ghorbani et al., 2022). Our code is available at `https://github.com/xqlin98/data-selection-efficient-topm`

**Baselines.** [3] (a) MC, the Monte-Carlo sampling approach (Castro et al., 2009; Ghorbani & Zou, 2019). (b) Owen, a multi-linear extension approach (Owen, 1972; Okhrati & Lipani, 2021). (c)

---

[3] We do not compare with Jia et al. (2019b) here as they do not publicize their codes. We provide a separate comparison with our implementation of Jia et al. (2019b) in Appendix B.

Sobol, a permutation sampling approach using the Sobol sequence (Mitchell et al., 2022). (d) Stratified, a stratified sampling approach (Castro et al., 2017). (e) KernelSHAP, a regression-based approach (Covert & Lee, 2021). (f) GapE, MAB algorithm from Kolpaczki et al. (2021). (g) BUS, MAB algorithm from Kolpaczki et al. (2021). (h) $m$-LinGapE, linear bandit algorithm (Réda et al., 2021). (i) GPGapE, our approach in Algorithm 1. (j) GPGapE-Small, GPGapE accelerated by querying marginal contributions on small subsets.

## 5.1 TOP-$m$ DATA VALUES IDENTIFICATION

**Game with easy-to-compute closed-form SV.** A game with easy-to-compute closed-form SV is needed to inspect the quality of top-$m$ data values identification for large datasets. Specifically, to inspect the quality of top-$m$ data values identification, we need the ground truth top-$m$ SV. However, knowing the exact top-$m$ SV for a dataset with a large number of data points (e.g., $10k$) is computationally infeasible since $n!$ of model training is required. We propose to define a game with a utility function specific to ML which enables us to derive an easy-to-compute closed-form SV. Denote $g : \mathcal{X} \times \mathcal{Y} \to \mathbb{R}^{d'}$ a function mapping a data point $z_i$ into a $d'$-dimensional latent space. Denote $D_V$ as the validation dataset. We define the following utility function:

$$U(S) := \frac{1}{|D_V|} \sum_{z_i \in D_V} \mathbb{1}\big(g(z_i) \in M(D_S, \varepsilon)\big), \quad M(D_S, \varepsilon) := \bigcup_{z_j \in D_S} \{z' \in \mathbb{R}^d : \rho(z', g(z_j)) \le \varepsilon\} \quad (7)$$

where $\mathbb{1}()$ is the indicator function and $\rho(\cdot, \cdot)$ is a distance measure. To elaborate, $M(D_S, \varepsilon)$ is the union of closed $\varepsilon$-balls defined by each data point in $D_S$. $U(S)$ is the fraction of data points (in the validation dataset) within the union of $\varepsilon$-balls formed by $D_S$. Intuitively, if $z_i$ in the validation dataset is within an $\varepsilon$-ball defined by a training data point $z_j$, the model trained on the dataset with $z_j$ is more likely to predict the label of $z_i$ correctly. Therefore, the utility function here means how well the training dataset can generalize to the validation dataset and is similar to "coverage" from existing active learning works (Joshi et al., 2012; Katragadda et al., 2022) which show that "coverage" is predictive of model performance.

Denote $D_V' = \{z_j \in D_V | \exists z_k \in D_N, \rho(g(z_k), g(z_j)) \le \varepsilon\}$. With the utility specified as Equ. (7):

$$\varphi_i = 1/|D_V| \sum_{z_j \in D_V'} \mathbb{1}\big(\rho(g(z_j), g(z_i)) \le \varepsilon\big) / |\{z_k \in D_N : \rho(g(z_k), g(z_j)) \le \varepsilon\}| . \quad (8)$$

This result is derived using the axioms of SV (see Appendix C). We can now obtain the exact SV efficiently (i.e., with a complexity of $O(|D_N||D_V|)$) to examine the quality of the top-$m$ data values identified by different approaches. Note that the design of this game is to benchmark the performance of different approaches. We also consider other scenarios without access to ground truth SV and use other evaluation metrics in Sec. 5.2 and Sec. 5.3.

We perform experiments on top-$m$ data values identification with closed-form SV. We use the MNIST dataset with $10k$ data points in the training dataset and $10k$ data points in the validation dataset. We train an NN with a three-layer multilayer perception (MLP). For a data point $z$, we use the last hidden layer representation of the NN as $g(z)$ in Equ. (7). We also use the same representation as the data point features used in our GP to update the GP posterior. We use the RBF kernel for GP and follow Lemma C.7 to set $C_{\delta,t} = 1 + \sqrt{(\ln t)^{d+1}}$ (i.e., the same scale as the theoretical $C_{\delta,t}$ w.r.t. $t$). Empirically, we update the GP posterior every 100 queries to marginal contributions to save computation. We set $p = \lfloor 0.1n \rfloor$ in GPGapE-small.

We use recall of ground truth top-$m$ data values $\mathcal{S}_m^*$ as the metric to evaluate the performance of different approaches. Fig. 2 shows that the recall of GPGapE increases quickly with only a few queries to the marginal contributions in the beginning while the recall for other approaches (e.g., KernelSHAP and Owen) improves very slowly. MC achieves better recall compared to KernelSHAP and Owen. Sobol achieves slightly better performance than MC. This is because Sobol is able to sample more diverse permutations, hence improving the sampling efficiency (Mitchell et al., 2022). Stratified sometimes performs better than Sobol while not in some others. Note that Stratified will focus on sampling different strata at different stages. When it samples the marginal contributions on large subsets, the data values approximation quality improves marginally, and vice versa (as we discussed in Sec. 4.2). Our GPGapE and GPGapE-Small perform better than all other baselines. Table 1 shows the query efficiency by different approaches to achieve the same recall, GPGapE-Small achieves the same recall as other baselines with $50\times$ fewer queries when $m = 1000$. GPGapE-Small performs better than GPGapE with smaller computational costs (see Table 1 and Table 2).

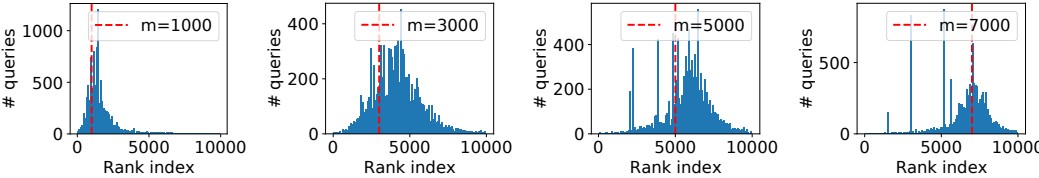

Figure 3: # queries for different data points of GPGapE. Data points are ranked by ground truth SV (high to low from left to right). Vertical dashed line is the position of the $m$-th largest SV.

Fig. 3 shows the frequency of being queried for different data points by our GPGapE. The data points around the ground-truth $m$-th valued data point are queried more frequently than other data points and the frequency drops when a data point is further away from the $m$-th valued data point in ranking. It means our algorithm does not waste the query budget on non-ambiguous data points as other approaches (e.g., MC). Instead, GPGapE samples marginal contributions around the ground truth $m$-th valued data point adaptively, thus achieving better query efficiency.

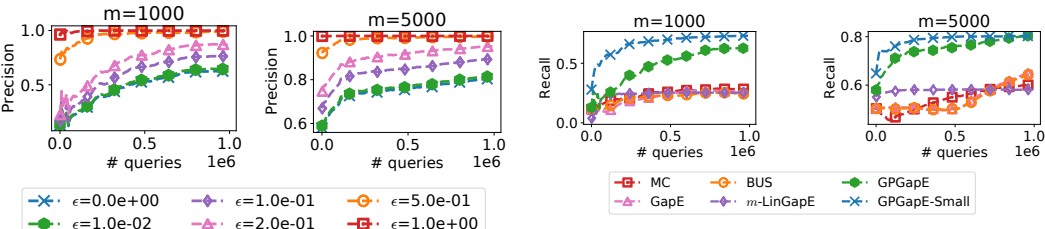

Figure 4: Precision under different $\varepsilon$ for GPGapE.

Figure 5: Recall of top-$m$ data values using GPGapE and other bandit algorithms.

Fig. 4 shows the precision of top-$m$ data values identified by GPGapE under different $\varepsilon$ where Precision $= |\hat{S}_m^* \cap S_m^{*,\varepsilon}|/|\hat{S}_m^*|$. When $\varepsilon$ is higher, GPGapE converges faster, aligning with our analysis in Theorem 4.2. Fig. 4 also shows that GPGapE can terminate with an $(\varepsilon, \delta)$-approximation in a finite step since the precision reaches 1.0.

**Comparison with other existing bandit algorithms.** We perform experiments on comparing GPGapE with existing top-$m$ arms identification algorithms. Fig. 5 shows that $m$-LinGapE outperforms GapE and BUS when few queries are made. This is because $m$-LinGapE uses arm features to model the mapping function from data features to data values while GapE and BUS do not (Réda et al., 2021). GPGapE performs significantly better than all existing bandit algorithms including $m$-LinGapE. This is because $m$-LinGapE assumes a linear relationship between arm features and rewards which is not applicable in data valuation. This is especially true when the dataset used is highly complex (e.g., image dataset here). Consequently, the data values approximated by $m$-LinGapE are not accurate which directly leads to poor performance. We provide an additional comparison with $m$-LinGapE in the simulated scenario in which the simulated mapping function is linear in Appendix B.

## 5.2 Noisy data detection

We perform noisy data detection on MNIST, FashionMNIST, and CIFAR10. Specifically, we select $3k$ data points from each dataset to perform top-$m$ identification. We consider the logistic regression and NN models. For NN, we use a two-layer MLP for MNIST and FashionMNIST, and a convolutional neural network (CNN) with two convolutional layers followed by a three-layer MLP. We specify $U(S)$ as the validation accuracy of the model trained on $D_S$ (Ghorbani & Zou, 2019; Wu et al., 2022). We select 500 data points in each dataset to add Gaussian noise $\mathcal{N}(0, 2)$ to the images as the noisy data. We identify lowest-$m$ data values (by taking the negative of marginal contributions in GPGapE) instead to detect noisy data since the noisy data are expected to have low data values. We use the recall of the noisy data points as the evaluation metric. From Fig. 6, GPGapE and GPGapE-small get the best performance on all datasets and models. Note that slow increase of recalls for other baselines

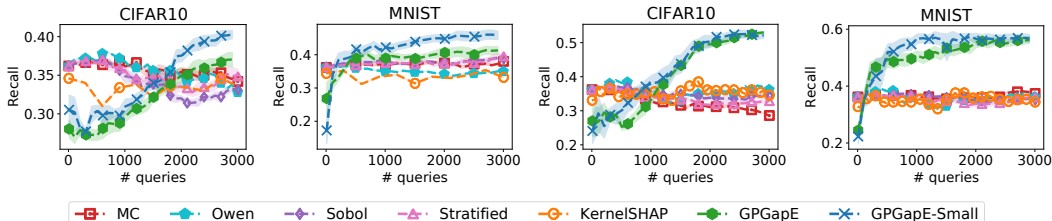

Figure 6: Recall of top-$m$ data values by different approaches in noisy data detection for logistic regression (left 2 figures) and NN (right 2 figures). More results in Appendix B.

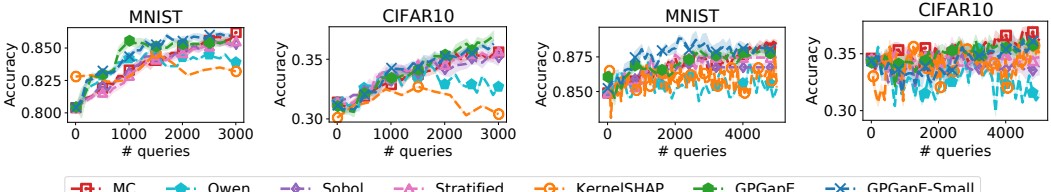

Figure 7: Validation accuracy of the data subset specified by top-$m$ data values by different approaches for logistic regression (left 2 figures) and NN (right 2 figures). More results in Appendix B.

makes their lines flat (see Fig. 13 with more queries). We update the GP posterior every 10 queries to save computation. We use principal component analysis to reduce the images to 32 dimensions as the data points features for logistic regression model and the last hidden representation of NN as the data points features for NN model. We set $p = \lfloor 0.3n \rfloor$ in GPGapE-small. More details in Appendix A.

## 5.3 DATA SUBSET SELECTION

We perform experiments using top-$m$ data values to select a data subset of size $m$, following the setting in Sec. 5.2. We use the validation accuracy achieved by data subsets specified by the top-$m$ data values from the different approaches as the evaluation metric. Fig. 7 shows that our GPGapE and GPGapE-Small perform the best among all approaches. To further validate the efficiency of our GPGapE, we perform data selection for Large Language Model (LLM) fine-tuning. Specifically, a previous work (Xia et al., 2024) proposes to use the LESS score to select a data subset that has better performance than the full dataset. We use our GPGapE to identify data points with top-$m$ LESS scores. We use the TYDIQA dataset (Clark et al., 2020) and LLAMA-2-7B model (Touvron et al., 2023) (more details in Appendix A). From Fig. 8, GPGapE

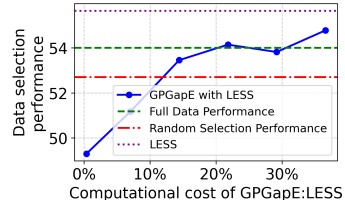

Figure 8: Performance of GPGapE in data selection for LLM fine-tuning.

uses 21.7% of the computation[4] required by LESS to find a data subset with the size of 5% of the full dataset that outperforms the full dataset. Moreover, GPGapE achieves performance comparable to LESS with only 36.6% of the computation cost of LESS, demonstrating its superior efficiency.

## 6 CONCLUSION AND LIMITATION

We propose the GPGapE algorithm for top-$m$ data values identification. We theoretically demonstrate the correctness of GPGapE and analyze its stopping iteration. Moreover, we exploit the diminishing return of ML models and hence propose to further accelerate our GPGapE by sampling marginal contributions on small data subsets. We empirically show the effectiveness of our GPGapE in top-$m$ data values identification, noisy data detection, and data subset selection on multiple real-world datasets. Of note, further improvements can still be made to our algorithm: 1) Better selection of data point features for GP in our GPGapE; 2) Approximate the utility function to further accelerate our algorithm. However, these are not the focus of our work and can be explored in future works.

---

[4]Computation is measured by the number of LESS scores used in GPGapE compared to LESS.

REPRODUCIBILITY STATEMENT

The source code for our experiments is included in the supplemental materials to ensure reproducibility. Details of datasets, computational resources, and hyper-parameters are provided in Appendix A.

ACKNOWLEDGEMENTS

This research is supported by the National Research Foundation Singapore and the Singapore Ministry of Digital Development and Innovation, National AI Group under the AI Visiting Professorship Programme (award number AIVP-2024-001). Xinyi Xu is supported by the Institute for Infocomm Research of Agency for Science, Technology and Research (A*STAR).

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

## A  ADDITIONAL DETAILS ON EXPERIMENT SETTINGS

### A.1  LICENSE FOR DATASETS

MNIST (LeCun et al., 1990): Attribution-Share Alike 3.0 License; CIFAR10 (Krizhevsky, 2009): MIT License; FashionMNIST (Xiao et al., 2017): MIT License.

### A.2  COMPUTATIONAL RESOURCES

Experiments are run on a server with AMD EPYC 7763 64-Core Processor, 1008GB RAM, and 8 NVIDIA L40 GPUs.

### A.3  ADDITIONAL DETAILS ON EXPERIMENTAL SETTINGS

**Data and model training.**  For training the logistic regression model, we apply principal component analysis (PCA) (Wold et al., 1987) on the training dataset to reduce the dimension to 32, hence reducing the running time of each logistic regression training. We randomly sample $1k$ data points as the validation dataset to further accelerate the utility evaluation (i.e., computation of model accuracy). Note that the same validation dataset is given to different approaches. The recall of noisy data detection is calculated based on the lowest-1000 data values identified by different approaches which is not the same as the ground truth 500. This is to simulate real-world scenarios when we do not have access to the number of noisy data points beforehand.

**Hyper-parameters.**  For logistic regression, we set the number of principal components as 32. For training MLP on MNIST, we set the learning rate to be $0.01$, and the number of epochs to be 10. For training MLP on FashionMNIST and CNN on CIFAR10, the learning rate is $0.001$, and the number of epochs is 30. We use a batch size of 200 and use Adam optimizer (Kingma & Ba, 2014) for all NN training. We use the RBF kernel as the kernel function $k$ for our GP. The length scale parameter of RBF is searched over $[0.5, 1, 10]$ and the noise parameter $\lambda = [1, 5, 10]$. We randomly select 100 data points to query one marginal contribution to initialize the GP posterior. Note that this is included in our query budget.

**Experimental setting for Fig. 1.**  To examine the diminishing return of machine learning model (Beleites et al., 2013; Mahajan et al., 2018), we compute the expected marginal contribution for data subsets of different sizes. Specifically, we perform our experiments on MNIST, CIFAR10, and FashionMNIST. For each dataset, we randomly select 25 data points from the original dataset to inspect their marginal contributions to different sizes of data subsets. For a fixed data subset size $l$, we randomly select 30 data subsets with size $l$ from the original dataset to approximate the expected marginal contribution for a data point to size $l$ data subsets. We range $l$ from $0 - 10k$ to obtain the plot in Fig. 1. The results are averaged over these 25 selected data points.

**Experimental setting for Fig. 8.**  We follow the same setting as Xia et al. (2024). Specifically, the data selection performance in Fig. 8 is the performance of the LLM trained on the data subset selected by different approaches. We use the LLAMA-2-7B model in our experiments. To use our GPGapE to approximate top-$m$ LESS scores, we use the LESS score as $\psi_i$ in our algorithm and use the data feature obtained from an embedding model. Specifically, for each data point in the training dataset (i.e., the fine-tuning dataset), we concatenate the prompt and response as the following template "Question: prompt, Answer: response" and use the Sentence-BERT (Reimers & Gurevych, 2019) as the embedding model[5] to obtain the data feature.

### A.4  ON THE RATIONALE OF CHOOSING GP TO MODEL THE DATA VALUE MAPPING FUNCTION

We provide theoretical and empirical justification on why similar data points will have similar Shapley values.

---

[5]We use the model from `https://huggingface.co/sentence-transformers/all-MiniLM-L6-v2`

**Theoretical justification.** The difference of Shapley value of $i$ and $j$ can be bounded by the distance of the data point $i$ and data point $j$ according to the following:

**Lemma A.1.** (Xu et al., 2024, Lemma 6) *For all $i, j \in N$, $\left(\forall S \subseteq N \setminus \{i, j\}\right)$ $\left|U(S \cup \{i\}) - U(S \cup \{j\})\right| \leq L d(i,j)\right) \implies |\psi_i - \psi_j| \leq ZL d(i,j)$ where $L \geq 0$ is a constant and $d(i,j)$ is some distance measure between $i$ and $j$, and $Z$ is the linear scaling parameter.*

The condition for the Lemma A.1 to hold is that if two data points $i$ and $j$ are similar in data space, the performance of dataset $S \cup \{i\}$ and $S \cup \{j\}$ should be similar. This should be trivially true since these two data points contribute similar information to the model (e.g., data from a specific subgroup in the data space) and hence the model performance should be similar.

**Empirical justification.** To give an empirical verification, we compute the distance of Shapley value and the distance of data feature for randomly sampled data point pairs from the dataset. Fig. 9 shows that when the distance of the data feature increases, the distance of Shapley values also increases, which validates the theoretical result above.

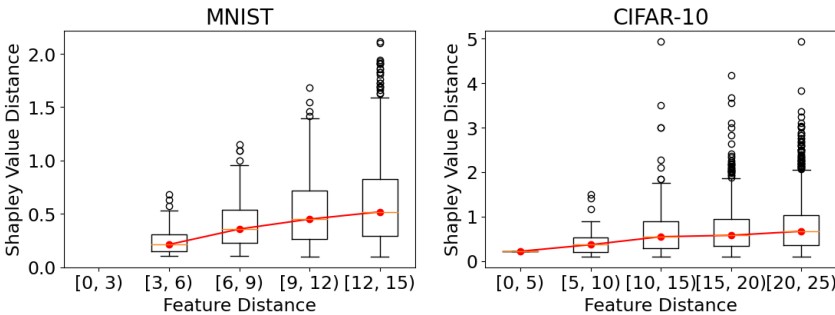

Figure 9: Shapley value distance and data feature distance for randomly selected data point pairs. $l - 2$ norm is used as the distance measure.

Since similar data points have similar Shapley values, GP is a good design choice for modeling the mapping function. Specifically, GP essentially uses the kernel-based method for prediction, and the idea of kernel function is based on the belief that similar input should have similar function output. Consequently, GP is a good choice for modeling the mapping function.

A.5 FAIRNESS PROPERTIES OF TOP-$m$ DATA VALUES

Top-$m$ data values are not only useful empirically in data subset selection and noisy data detection. We show that the resulting data values defined by the identified top-$m$ data values inherit several fairness properties of the exact data values. We define the data values with the exact top-$m$ data values $\mathcal{S}_m^*$ as:

$$\varphi_i^{(m)} = \begin{cases} U(N)/m, & \text{if } i \in \mathcal{S}_m^* \\ 0, & \text{if } i \notin \mathcal{S}_m^* \end{cases} \tag{9}$$

**Proposition A.2** (Fairness properties of $\varphi_i^{(m)}$). *Assume that we can arrange $\varphi_1 \geq \varphi_2 \geq \cdots \geq \varphi_m > \varphi_{m+1} \geq \varphi_{m+2} \geq \cdots \geq \varphi_n$ for $m \in \{1, \ldots, n-1\}$ (i.e., assuming the uniqueness of the top-$m$ data values) and $U(N) > 0$. $\varphi_i^{(m)}$ satisfies the following fairness properties:*

- **Efficiency.** $\sum_{i \in N} \varphi_i^{(m)} = U(N)$.

- **Symmetry.** $\left(\forall S \in N \setminus \{i, j\}, U(S \cup \{i\}) = U(S \cup \{j\})\right) \implies \varphi_i^{(m)} = \varphi_j^{(m)}$.

- **Strict $m$-th desirability.** $\left(\exists B \in N \setminus \{i, j\}, U(B \cup \{i\}) > U(B \cup \{j\})\right) \wedge \left(\forall C \in N \setminus \{i, j\}, U(B \cup \{i\}) \geq U(B \cup \{j\}) \wedge (\varphi_j = \varphi_{m+1})\right) \implies \varphi_i^{(m)} > \varphi_j^{(m)}$.

To elaborate, efficiency means that the utility obtained by the grand coalition $N$ will be all allocated to different participants (i.e., data points in our case). Symmetry means that two data points that

always have the same utility when added to different subsets $S$ will get the same data value. Strict $m$-th desirability means that if a data point contributes strictly more to a subset $S$ than the data points whose original data value $\varphi$ is equal to the $m + 1$-th data value and contribute no less in other subsets, then it will receive strictly better value than data point with original data value as $\varphi_{m+1}$. These fairness properties are useful for incentive mechanism designs in collaborative machine learning (Sim et al., 2020; Tay et al., 2022).

## B  ADDITIONAL EXPERIMENTAL RESULTS

| $m$ | Best recall by other approaches | # queries other approaches | # queries GPGapE | Speedup: GPGapE compared to other approaches | # queries GPGapE-Small | Speedup: GPGapE-Small compared to other approaches (compared to GPGapE) |
|---|---|---|---|---|---|---|
| 1000 | 0.441 | 1000k | 330k | **3.03×** | 20k | **50.00×(16.50×)** |
| 3000 | 0.625 | 1000k | 220k | **4.55×** | 40k | **25.00×(5.50×)** |
| 5000 | 0.750 | 1000k | 380k | **2.63×** | 100k | **10.00×(3.80×)** |
| 7000 | 0.867 | 1000k | 230k | **4.35×** | 80k | **12.50×(2.88×)** |

Table 1: Speedup in # of queries by our GPGapE and GPGapE-Small compared with other approaches. The best recall in the second column is the best recall achieved by other approaches in $1000k$ queries. The # queries show the number of queries that different approaches achieve this recall in the second column, the lower the better. The speedup of GPGapE and GPGapE-Small is compared with other approaches, the higher the better.

Table 1 shows the speedup of our GPGapE and GPGapE-Small when compared to other existing data value approximation approaches. The setting is exactly the same as Fig. 2. We can see that GPGapE-Small is $50\times$ better in query efficiency than other existing data value approximation approaches when $m = 1000$ and is $16.5\times$ better than GPGapE in query efficiency when $m = 1000$.

| Dataset | Running time of GPGapE (mins) | Running time of GPGapE-Small (mins) | Speedup |
|---|---|---|---|
| MNIST | 8.13(0.53) | 4.89(0.48) | 1.66× |
| CIFAR10 | 35.44(1.55) | 18.53(0.41) | 1.91× |

Table 2: Speedup in the running time of our GPGapE-Small compared with GPGapE.

Table 2 shows the actual running time of GPGapE and GPGapE-Small on MNIST and CIFAR10 when NN is used when the same number of queries are used. Compared to GPGapE, GPGapE-Small is $1.91\times$ faster than GPGapE. This speedup will be more significant when $p$ is set to a smaller value and when the dataset or model is larger.

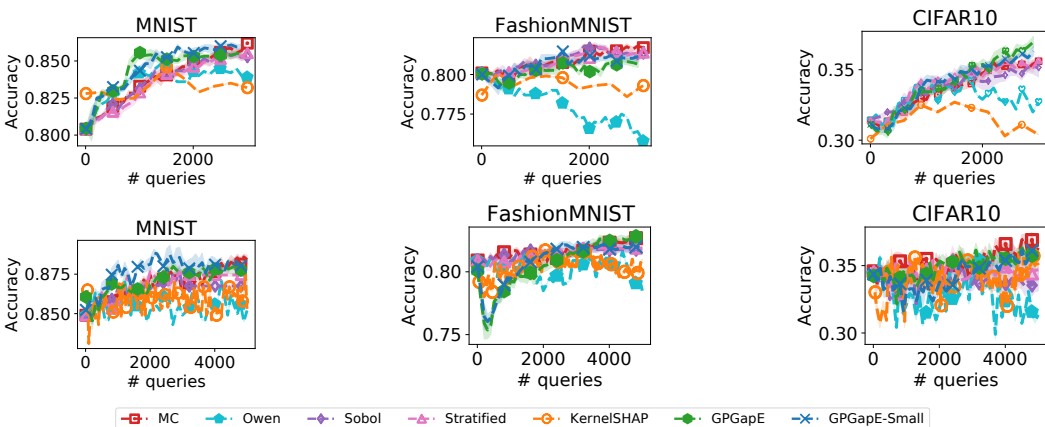

Figure 10: Validation accuracy of the data subset specified by top-$m$ data values by different approaches for logistic regression (row 1) and NN (row 2).

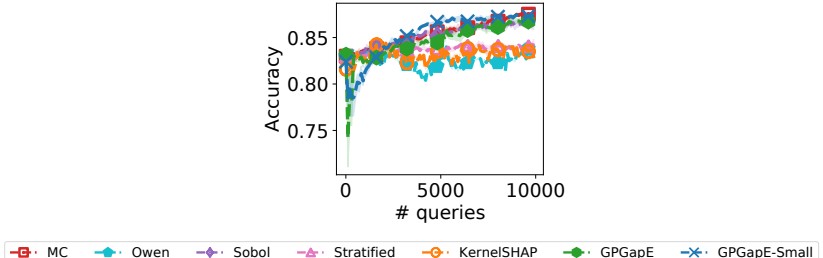

Figure 11: Validation accuracy of the data subset specified by top-$m$ data values by different approaches. The number of data points to be evaluated is $10k$.

Fig. 10 is a more complete version of Fig. 7 for data subset selection since it includes the results for all datasets. Fig. 11 is the data subset selection results on a larger dataset with $10k$ data points for logistic regression. Our GPGapE-Small still performs the best among all approaches.

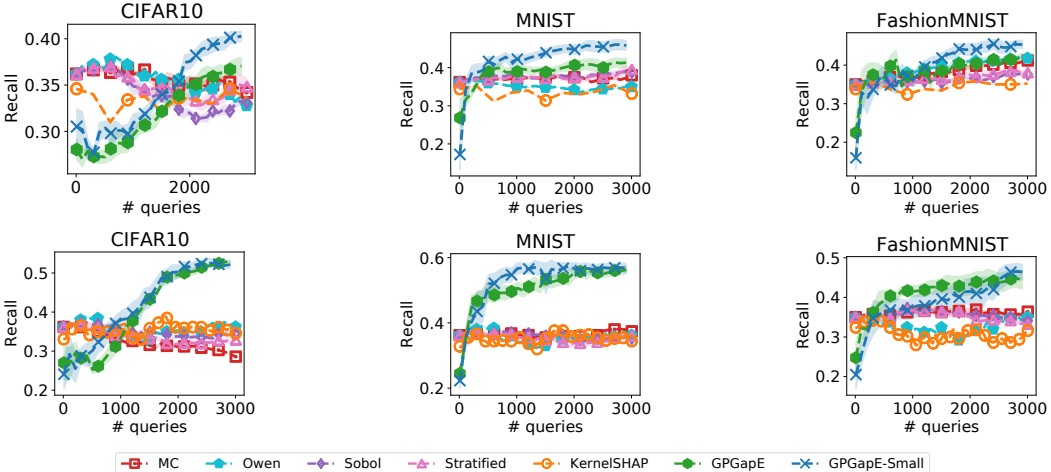

Figure 12: Recall of top-$m$ data values by different approaches in noisy data detection for logistic regression (row 1) and NN (row 2).

Fig. 12 is a more complete version of Fig. 6 for noisy data detection since it includes the results for all datasets.

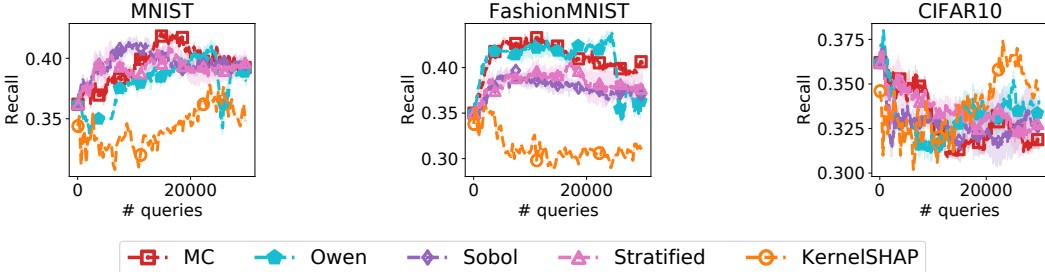

Figure 13: Recall of top-$m$ data values using different approximation approaches in noisy data detection with $30k$ number of queries.

Fig. 13 is the result for the same setting as Fig. 6, except that it shows more iterations. As we can see that the performance for other existing data value approximation approaches increases very slowly compared to our GPGapE in Fig. 6.

**Additional comparison with Jia et al. (2019b).** We provide an additional comparison with the group testing method (GroupTest) proposed in Jia et al. (2019b). Since the code of GroupTest is not

released, especially the solver used to solve the linear programming in GroupTests is not provided, we use the most used solver provided by scipy library (i.e., scipy.optimize.linprog). However, since there is $n(n-1)$ number of constraints, solving the programming for $n = 10k$ (i.e. almost 100 million constraints) takes a lot of time (i.e., 7 hours for the solving step alone), therefore, we are unable to provide the progressive result in Fig. 2 which requires solving the programming after every new query. Therefore, we provide the performance (i.e., recall) after all the queries are done in Table 3.

| m | GroupTest | GPGapE |
|------|-----------|--------|
| 1000 | 0.1170    | **0.2150** |
| 3000 | 0.2967    | **0.5130** |
| 5000 | 0.4940    | **0.6340** |

Table 3: Recall of top-$m$ data values identified by GroupTest and GPGapE after 100000 queries for $n = 10k$ on MNIST dataset.

To also provide a progressive result as Fig. 2, we perform experiments on $n = 1000$ (since the running time of the solver is manageable when $n = 1000$), the result is in Fig. 14.

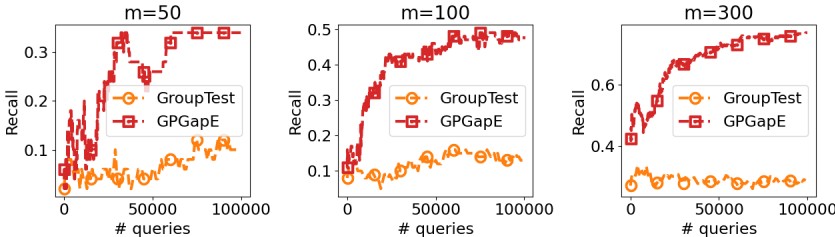

Figure 14: Comparison of our GPGapE and the GroupTest method in Jia et al. (2019b) on MNIST dataset. The number of data points in the dataset is $1000$.

**Additional comparison with $m$-LinGapE algorithm.** We provide an additional comparison with $m$-LinGapE in the simulated scenario in which the function mapping from data feature to data values is simulated to be a linear function. In this case, GPGapE is expected to perform similarly to $m$-LinGapE (which assumes linearity). This is because when GP uses the linear kernel function (i.e., $k(x_1, x_2) = x_1^T x_2$), GP posterior mean is exactly the closed-form of linear regression.

We use a linear function $y = \theta^T x$ where $x, \theta \in \mathbb{R}^{10}$ to simulate the ground truth mapping function. The number of data points $x$ is $10k$. Both $x$ and $\theta$ are randomly sampled from the standard Gaussian. A noise randomly sampled from $\mathcal{N}(0, 1e-4)$ is used to simulate the noisy observation. The result in Fig. 15 shows that GPGapE performs similarly to $m$-LinGapE.

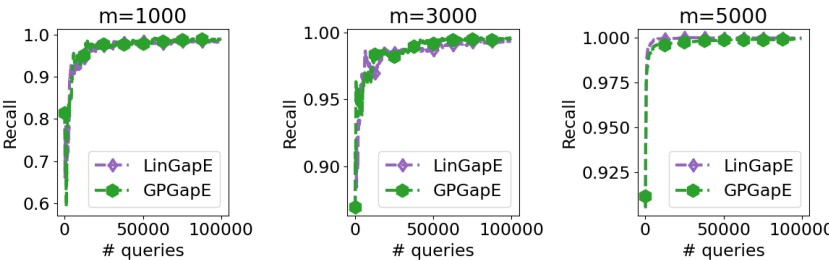

Figure 15: Comparison of our GPGapE and the $m$-LinGapE when the mapping function is a linear function.

Note that in real data valuation scenarios, the linearity between the data features and the data values usually does not hold and hence our GPGapE is able to outperform $m$-LinGapE in our top-$m$ data values identification experiments.

**Additional investigation of the diminishing return for top-$m$ data points.** To investigate whether the diminishing return is applicable for the data points with top-$m$ data values, we plot the $|\Delta_i^l|$ w.r.t. different sizes of data subset $l$ for top-10 data points in the MNIST dataset in Fig. 16. The result is consistent with Fig. 1. Note that for each individual point (Fig. 16 left), even though $|\Delta_i^l|$ is not monotonically decreasing, the general trend is decreasing with minor ups and downs, which still supports our assumption of $\exists p \in N$ such that $|\Delta_i^l| \leq \varepsilon', \forall l \in \{p, \ldots, n\}$ for Prop. 4.5.

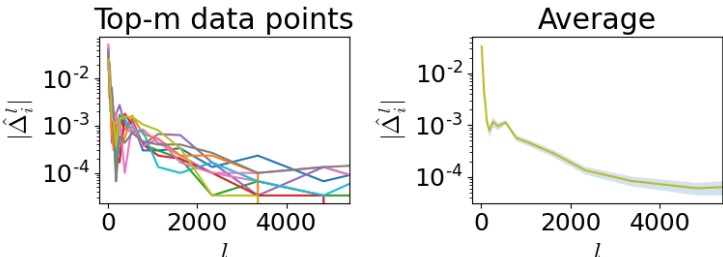

Figure 16: Diminishing return of adding a top-$m$ data point to the data subset when the size of the data subset increases. The result for each individual top-$m$ data point is on the left and the result for the average (over top-$m$ data points) is on the right. Marginal contributions are computed via the validation accuracy (details in Appendix A).

## C   PROOFS FOR THE THEORETICAL RESULTS

### C.1   PROOF FOR THEOREM 4.1

Our proof requires the following results from the existing work (Réda et al., 2021):

**Definition C.1.** (Réda et al., 2021, Definition 1) Let us denote

$$\xi_m := \bigcap_{t>0} \bigcap_{j \in (\mathcal{S}_m^{*,\varepsilon})^c} \bigcap_{k \in \mathcal{S}_m^*} (B_t(z_k, z_j) \geq \varphi_k - \varphi_j).$$

A good choice of gap indices $\{B_t(z_i, z_j)\}_{i,j \in N, t>0}$ satisfies $\mathbb{P}(\xi_m) \geq 1 - \delta$.

**Lemma C.2.** (Réda et al., 2021, Theorem 1) On the event $\xi_m$ defined in Definition C.1, when the stopping condition $B_t(z_{c_t}, z_{b_t}) \leq \varepsilon$ holds with $b_t = \arg\max_{j \in J(t)} \max_{i \notin J(t)} B_{t-1}(z_i, z_j)$ and $c_t = \arg\max_{a \notin J(t)} B_{t-1}(z_a, z_{b_t})$, $\hat{S}_m^{\tau_\delta} \subseteq \mathcal{S}_m^{*,\varepsilon}$.

We need to proof the following lemma first:

**Lemma C.3.** Assume that $\{\eta_t\}_{t=1}^\infty$ are $R$-sub-Gaussian. Let $k(\cdot, \cdot)$ be a positive-semidefinite kernel function and let $\delta \in (0, 1]$. Assume that $f$ is a member of the reproducing kernel Hilbert space (RKHS) $H$ corresponding to the kernel function $k$ with RKHS norm bounded by $B$. The $\{B_t(z_i, z_j)\}_{i,j \in N, t>0}$ defined in Equ. (4) is a good choice of gap indices (i.e., $\mathbb{P}(\xi_m) \geq 1 - \delta$) when the noise parameter $\lambda$ is set to be $1 + 2/\tau_\delta$ and

$$C_{\delta,t} = B + R\sqrt{2(\gamma_t + 1 + \ln(1/\delta))}.$$

*Proof.* The proof is inspired by (Chowdhury & Gopalan, 2017). Define $\psi(z)$ as a mapping function where $\psi : \mathcal{X} \times \mathcal{Y} \to H$ maps any data point $z$ to the RKHS associated with $k$. For any two members $g, h \in H$, define the inner product $\langle g, h \rangle_k$ as $g^T h$ and the RHKS norm $\|g\|_k$ as $\sqrt{g^T g}$. Since $f$ is a member of $H$, we can write $f(z) = \langle f, \psi(z) \rangle_k = f^T \psi(z)$. Define $\Psi_t := [\psi_{(z_1)}^T, \ldots, \psi_{(z_t)}^T]^T$. We have that the kernel matrix is $K_t = \Psi_t \Psi_t^T, k_t(z) = \Psi_t \psi(z)$. Since $(\Psi_t^T \Psi_t + \lambda I)\Psi_t^T = \Psi_t^T(\Psi_t \Psi_t^T + \lambda I)$ and they are both strictly positive definite, we have $\Psi_t^T(\Psi_t \Psi_t^T + \lambda I)^{-1} = (\Psi_t^T \Psi_t + \lambda I)^{-1}\Psi_t^T$. From the definition, we have $(\Psi_t^T \Psi_t + \lambda I)\psi(z) = \Psi_t^T k_t(z) + \lambda \psi(z)$. Hence we have

$$\psi(z) = (\Psi_t^T \Psi_t + \lambda I)^{-1}\Psi_t^T k_t(z) + \lambda(\Psi_t^T \Psi_t + \lambda I)^{-1}\psi(z),$$

which gives

$$\psi(z)^T \psi(z) = k_t(z)^T(\Psi_t \Psi_t^T + \lambda I)^{-1}k_t(z) + \lambda \psi(z)^T(\Psi_t^T \Psi_t + \lambda I)^{-1}\psi(z).$$

This will give us
$$\lambda\psi(z)^T(\Psi_t^T\Psi_t+\lambda I)^{-1}\psi(z)=k(z,z)-k_t(z)^T(\Psi_t\Psi_t^T+\lambda I)^{-1}k_t(z)=\sigma_t^2(z)\ .$$
Similarly, we have
$$\lambda\psi(z_i)^T(\Psi_t^T\Psi_t+\lambda I)^{-1}\psi(z_j)=k(z_i,z_j)-k_t(z_i)^T(\Psi_t\Psi_t^T+\lambda I)^{-1}k_t(z_j)=k_t^2(z_i,z_j)\ .$$
We observe that
$$\begin{aligned}f(z)-k_t(z)^T(K_t+\lambda I)^{-1}f_{1:t}&=\psi(z)^Tf-\psi(z)^T\Psi_t^T(\Psi_t\Psi_t^T+\lambda I)^{-1}\Psi_tf\\&=\psi(z)^Tf-\psi(z)^T(\Psi_t^T\Psi_t+\lambda I)^{-1}\Psi_t^T\Psi_tf\\&=\lambda\psi(z)^T(\Psi_t^T\Psi_t+\lambda I)^{-1}f\ .\end{aligned}$$
Hence we have that
$$\begin{aligned}&\left|\left(f(z_i)-k_t(z_i)^T(K_t+\lambda I)^{-1}f_{1:t}\right)-\left(f(z_j)-k_t(z_j)^T(K_t+\lambda I)^{-1}f_{1:t}\right)\right|\\&=|\lambda(\psi(z_i)-\psi(z_j))^T(\Psi_t^T\Psi_t+\lambda I)^{-1}f|\\&\le\|\lambda(\Psi_t^T\Psi_t+\lambda I)^{-1}(\psi(z_i)-\psi(z_j))\|_k\|f\|_k\\&=\|f\|_k\sqrt{\lambda(\psi(z_i)-\psi(z_j))^T(\Psi_t^T\Psi_t+\lambda I)^{-1}\lambda I(\Psi_t^T\Psi_t+\lambda I)^{-1}(\psi(z_i)-\psi(z_j))}\\&\le B\sqrt{\lambda(\psi(z_i)-\psi(z_j))^T(\Psi_t^T\Psi_t+\lambda I)^{-1}(\Psi_t^T\Psi_t+\lambda I)(\Psi_t^T\Psi_t+\lambda I)^{-1}(\psi(z_i)-\psi(z_j))}\\&=B\sqrt{\lambda(\psi(z_i)-\psi(z_j))^T(\Psi_t^T\Psi_t+\lambda I)^{-1}(\psi(z_i)-\psi(z_j))}\\&=B\sqrt{\lambda\psi(z_i)^T(\Psi_t^T\Psi_t+\lambda I)^{-1}\psi(z_i)+\lambda\psi(z_j)^T(\Psi_t^T\Psi_t+\lambda I)^{-1}\psi(z_j)-2\lambda\psi(z_i)^T(\Psi_t^T\Psi_t+\lambda I)^{-1}\psi(z_j)}\\&=B\sqrt{\sigma_t^2(z_i)+\sigma_t^2(z_j)-2k_t(z_i,z_j)}\ .\end{aligned}$$
Furthermore, we have that
$$\begin{aligned}&|k_t(z_i)^T(K_t+\lambda I)^{-1}\eta_{1:t}-k_t(z_i)^T(K_t+\lambda I)^{-1}\eta_{1:t}|\\&=|(\psi(z_i)-\psi(z_j))^T\Psi_t^T(\Psi_t\Psi_t^T+\lambda I)^{-1}\eta_{1:t}|\\&=|(\psi(z_i)-\psi(z_j))^T(\Psi_t^T\Psi_t+\lambda I)^{-1}\Psi_t^T\eta_{1:t}|\\&\le\left\|(\Psi_t^T\Psi_t+\lambda I)^{-1/2}(\psi(z_i)-\psi(z_j))\right\|_k\left\|(\Psi_t^T\Psi_t+\lambda I)^{-1/2}\Psi_t^T\eta_{1:t}\right\|_k\\&=\sqrt{(\psi(z_i)-\psi(z_j))^T(\Psi_t^T\Psi_t+\lambda I)^{-1}(\psi(z_i)-\psi(z_j))}\sqrt{(\Psi_t^T\eta_{1:t})^T(\Psi_t^T\Psi_t+\lambda I)^{-1}\Psi_t^T\eta_{1:t}}\\&=\lambda^{-1/2}\sqrt{\sigma_t^2(z_i)+\sigma_t^2(z_j)-2k_t(z_i,z_j)}\sqrt{\eta_{1:t}\Psi_t\Psi_t^T(\Psi_t\Psi_t^T+\lambda I)^{-1}\eta_{1:t}}\\&=\lambda^{-1/2}\sqrt{\sigma_t^2(z_i)+\sigma_t^2(z_j)-2k_t(z_i,z_j)}\sqrt{\eta_{1:t}K_t(K_t+\lambda I)^{-1}\eta_{1:t}}\ .\end{aligned}$$
According to the previous two inequality derived with previous definition of $y_t'=f(z_t)+\eta_t$ in Sec. 2, we have
$$\begin{aligned}&|\left(\mu_t(z_i)-\mu_t(z_j)\right)-(f(z_i)-f(z_j))|\\&=|\left(k_t(z_i)^T(K_t+\lambda I)^{-1}(f_{1:t}+\eta_{1:t})-k_t(z_j)^T(K_t+\lambda I)^{-1}(f_{1:t}+\eta_{1:t})\right)-(f(z_i)-f(z_j))|\\&\le|\left(k_t(z_i)^T(K_t+\lambda I)^{-1}f_{1:t}-k_t(z_j)^T(K_t+\lambda I)^{-1}f_{1:t}\right)-(f(z_i)-f(z_j))|\\&\quad+|k_t(z_i)^T(K_t+\lambda I)^{-1}\eta_{1:t}-k_t(z_j)^T(K_t+\lambda I)^{-1}\eta_{1:t}|\\&=|\left(f(z_i)-k_t(z_i)^T(K_t+\lambda I)^{-1}f_{1:t}\right)-\left(f(z_j)-k_t(z_j)^T(K_t+\lambda I)^{-1}f_{1:t}\right)|\\&\quad+|k_t(z_i)^T(K_t+\lambda I)^{-1}\eta_{1:t}-k_t(z_j)^T(K_t+\lambda I)^{-1}\eta_{1:t}|\\&\le\left(B+\lambda^{-1/2}\sqrt{\eta_{1:t}K_t(K_t+\lambda I)^{-1}\eta_{1:t}}\right)\sqrt{\sigma_t^2(z_i)+\sigma_t^2(z_j)-2k_t(z_i,z_j)}\ .\end{aligned}$$
Let $\lambda=1+\omega$ where $\omega>0$. Let $K=K_t+\omega I$ and hence $K$ is reversible, $K(K+I)^{-1}=((K+I)K^{-1})^{-1}=(I+K^{-1})^{-1}$. Replacing $K=K_t+\omega I$, we have
$$(K_t+\omega I)(K_t+(1+\omega)I)^{-1}=((K_t+\omega I)^{-1}+I)^{-1}\ .$$

By using the above equation, We have

$$\eta_{1:t} K_t (K_t + \lambda I)^{-1} \eta_{1:t} \leq \eta_{1:t} (K_t + \omega I)(K_t + (1+\omega)I)^{-1} \eta_{1:t} = \eta_{1:t} ((K_t + \omega I)^{-1} + I)^{-1} \eta_{1:t}$$

Using Theorem 1 from Chowdhury & Gopalan (2017), with probability at least $1 - \delta, \forall t > 0, \forall z \in \mathcal{X} \times \mathcal{Y}$, we have

$$\sqrt{\eta_{1:t}((K_t + \omega I)^{-1} + I)^{-1}\eta_{1:t}} \leq R\sqrt{2\ln\frac{\sqrt{\det((1+\omega)I + K_t)}}{\delta}}$$

$$= R\sqrt{2\ln\frac{\sqrt{\det(I + (1+\omega)^{-1}K_t)\det((1+\omega)I)}}{\delta}}$$

$$= R\sqrt{\ln(\det((1+\omega)I + K_t)) + t\ln(1+\omega) + 2\ln(1/\delta)}$$

$$\leq R\sqrt{2\gamma_t + \omega t + 2\ln(1/\delta)} \,.$$

We choose a small $\omega = 2/\tau_\delta$ where $\tau_\delta$ is the termination iteration for the algorithm. Hence we get

$$\sqrt{\eta_{1:t}((K_t + \omega I)^{-1} + I)^{-1}\eta_{1:t}} \leq R\sqrt{2(\gamma_t + 1 + \ln(1/\delta))} \,.$$

Therefore, we have that with probability at least $1 - \delta, \forall t > 0, \forall z \in \mathcal{X} \times \mathcal{Y}$, we have

$$\left|\left(\mu_t(z_i) - \mu_t(z_j)\right) - \left(f(z_i) - f(z_j)\right)\right| \leq \left(B + R\sqrt{2(\gamma_t + 1 + \ln(1/\delta))}\right)\sqrt{\sigma_t^2(z_i) + \sigma_t^2(z_j) - 2k_t(z_i, z_j)} \,.$$

Rearrange the above equation, we get:

$$\varphi_i - \varphi_j \leq \mu_t(z_i) - \mu_t(z_j) + \left(B + R\sqrt{2(\gamma_t + 1 + \ln(1/\delta))}\right)\sqrt{\sigma_t^2(z_i) + \sigma_t^2(z_j) - 2k_t(z_i, z_j)} = B_t(z_i, z_j)$$

where $B_t(z_i, z_j)$ is defined with $C_{\delta,t} = \left(B + R\sqrt{2(\gamma_t + 1 + \ln(1/\delta))}\right)$. According to Definition C.1, $B_t(z_i, z_j)$ is a good choice of gap indices. □

*Proof of Theorem 4.1.* Combining Lemma C.3 and Lemma C.2, we have that the output by GPGapE $\hat{S}_m^{\tau_\delta} \in \mathcal{S}_m^{*,\varepsilon}$ with probability at least $1 - \delta$. □

## C.2 PROOF FOR THEOREM 4.2

We need the following results from the existing work (Réda et al., 2021):

**Lemma C.4.** (Réda et al., 2021, Lemma 4) In Algorithm 1, for any selection rule, on event $\xi := \bigcap_{t>0} \bigcap_{i,j \in N} (G(z_i, z_j) \in [-B_t(z_i, z_j), B_t(z_i, z_j)])$, with the form of $B_t(z_i, z_j) = \hat{G}_t(z_i, z_j) + W_t(z_i, z_j)$, *for all* $t > 0$, $B_t(z_{c_t}, z_{b_t}) \leq \min(-\max(G_{b_t}, G_{c_t}) + 2W_t(z_{b_t}, z_{c_t}), 0) + W_t(z_{b_t}, z_{c_t})$ .

**Lemma C.5.** (Réda et al., 2021, Lemma 6) Let $T^* : N \times (0, 1) \times \mathbb{N}^* \to \mathbb{R}^+$ be a function that is nondecreasing in $t$, and $\mathcal{I}_t$ is the set of pulled arms at time $t$. Let $\xi$ be an event that for all $t < \tau_\delta, \delta \in (0, 1), \exists a_t \in \mathcal{I}_t$ the number of arm pulls $N_t(a_t)$ at time $t$ for the arm $a_t$ satisfies $N_t(a_t) \leq T^*(a_t, \delta, t)$. Then it holds on the event $\xi$ that $\tau_\delta \leq T(\mu, \delta)$ where

$$T(\mu, \delta) := \inf\{u \in \mathbb{R}^+ : u > 1 + \sum_{a \in N} T^*(a, \delta, u)\} \,.$$

We need to proof the following lemma first:

**Lemma C.6.** $\forall t > 0, \tau_\delta > t, N_t(a_t) \leq T^*(a_t, \delta, t)$, where $a_t \in N$ is the index of a queried point at time $t$ and $N_t(a_t)$ is number of queries done for $a_t$ during time $1 : t$, and

$$T^*(a_t, \delta, t) = 12C_{\delta,t}^2 \max(\varepsilon, \frac{\varepsilon + G_{a_t}}{3})^{-2} \,.$$

*Proof.* Since at the stopping iteration $\tau_\delta$ we have

$$\varepsilon \leq B_t(z_{c_t}, z_{b_t}) \leq \min(-\max(G_{b_t}, G_{c_t}) + 3W_t(z_{b_t}, z_{c_t}), W_t(z_{b_t}, z_{c_t})) \,.$$

The inequality above is from Lemma C.4. Hence we have

$$\max(\varepsilon, \frac{\varepsilon + G_{b_t}}{3}, \frac{\varepsilon + G_{c_t}}{3}) \leq W_t(z_{b_t}, z_{c_t}) = C_{\delta,t}\sqrt{\sigma_t^2(z_{b_t}) + \sigma_t^2(z_{c_t}) - 2k_t(z_{c_t}, z_{b_t})}$$
$$\leq 2C_{\delta,t}\sigma_t(z_{a_t})$$
$$(\text{where } a_t = \max_{a \in \{b_t, c_t\}} \sigma_t(z_a))$$
$$= 2C_{\delta,t}\sqrt{\lambda\psi(z_{a_t})^T(\Psi_t^T\Psi_t + \lambda I)^{-1}\psi(z_{a_t})}$$
$$= 2\lambda^{1/2}C_{\delta,t}\|\psi(z_{a_t})\|_{(\Psi_t^T\Psi_t + \lambda I)^{-1}}$$
$$= 2\lambda^{1/2}C_{\delta,t}\|\psi(z_{a_t})\|_{(\sum_{a \in N} N_t(a)\psi(z_a)\psi(z_a)^T + \lambda I)^{-1}}$$
$$\leq 2\lambda^{1/2}C_{\delta,t}\frac{\|\psi(z_{a_t})\|_k}{\sqrt{N_t(a_t)}\|\psi(z_{a_t})\|_k}$$
$$= 2\lambda^{1/2}C_{\delta,t}\frac{1}{\sqrt{N_t(a_t)}} .$$

Hence we have,

$$N_t(a_t) \leq \frac{4\lambda C_{\delta,t}^2}{\max(\varepsilon, \frac{\varepsilon + G_{a_t}}{3})^2}$$
$$\leq \frac{12 C_{\delta,t}^2}{\max(\varepsilon, \frac{\varepsilon + G_{a_t}}{3})^2} = T^*(a_t, \delta, t) .$$

The last inequality is because $\lambda = 1 + 2/\tau_\delta \leq 3$ (see the proof for Theorem 4.1). $\qquad\square$

*Proof of Theorem 4.2.* Combining Lemma C.5 and Lemma C.6, we get the result that

$$\tau_\delta \leq \inf\{u \in \mathbb{R}^+ : u > 1 + 12 \sum_{a \in N} \max(\varepsilon, \frac{\varepsilon + G_a}{3})^{-2} C_{\delta,u}^2\} . \qquad (10)$$

$\qquad\square$

## C.3 PROOF FOR PROPOSITION 4.3

Our proof relies on the following results:

**Lemma C.7.** (Srinivas et al., 2010, Theorem 5) Let $D \subset \mathbb{R}^d$ be compact and convex, denote the dimension of $z$ as $d \in \mathbb{N}$. Assume that the kernel function satisfies $\forall z, z', k(z, z') \leq 1$.

- If $k$ is the linear kernel function: $\gamma_t = \mathcal{O}(d \log t)$.

- If $k$ is the RBF kernel function: $\gamma_t = \mathcal{O}((\log t)^{d+1})$.

**Lemma C.8.** (Chatzigeorgiou, 2013, Theorem 1) The Lambert function $W_{-1}(-e^{-x-1})$ for $x > 0$ is bounded as follows

$$-1 - \sqrt{2x} - x < W(-e^{-x-1}) < -1 - \sqrt{2x} - \frac{2}{3}x .$$

*Proof of Proposition 4.3.* When $k$ is the RBF kernel function, since $\gamma_t = \mathcal{O}((\log t)^{d+1})$ according to Lemma C.7, we can find an $c_0$ and a $t_0$ such that when $t \geq t_0$, $\gamma_t \leq c_0(\ln t)^{d+1}$. We have:

$$C_{\delta,t}^2 = (B + R\sqrt{2(\gamma_t + 1 + \ln(1/\delta))})^2$$
$$\leq (B + R\sqrt{2((c_0 \ln t)^{d+1} + 1 + \ln(1/\delta))})^2$$

where recall that $B$ is the upper bound for the norm of $f$ and $R$ is the parameter for the sub-Gaussian. Denote $M := \max_{a \in N}(\max(\varepsilon, \frac{\varepsilon + G_a}{3})^{-2})$. From Equ. (10), we have:

$$
\begin{aligned}
\tau_\delta &\leq \inf\{u \in \mathbb{R}^+ : u > 1 + 12 \sum_{a \in N} \max(\varepsilon, \frac{\varepsilon + G_a}{3})^{-2} C_{\delta,u}^2\} \\
&\leq \inf\{u \in \mathbb{R}^+ : u > 1 + 12n C_{\delta,u}^2 M\} \\
&\leq \inf\{u \in \mathbb{R}^+ : u > \underbrace{1 + 12n\Big(B + R\sqrt{2\big((c_0 \ln u)^{d+1} + 1 + \ln(1/\delta)\big)}\Big)^2 M}_{\mathcal{O}(n(\ln u)^{d+1})}\}.
\end{aligned}
$$

We can see that the right-hand side of the inequality in the brackets is $\mathcal{O}(n(\ln u)^{d+1})$. Therefore, there exists a $c_1 > 0$ and a $t_1 > 0$ such that when $u \geq t_1$, we have the right-hand side of the inequality in the brackets is no larger than $c_1 n(\ln u)^{d+1}$. Therefore, we have:

$$
\begin{aligned}
\tau_\delta &\leq \inf\{u \in \mathbb{R}^+ : u > c_1 n(\ln u)^{d+1}\} \\
&= \inf\{u \in \mathbb{R}^+ : \frac{u}{(\ln u)^{d+1}} > c_1 n\}
\end{aligned}
\tag{11}
$$

Let a function $h(u) = u/(\ln u)^{d+1}$. We have:

$$
\frac{\partial h(u)}{\partial u} = \frac{\ln(u) - (d+1)}{\ln^{d+2}(u)} .
$$

Therefore, when $u < e^{d+1}$, h(u) is monotonically decreasing. When $u \geq e^{d+1}$, $h(u)$ is monotonically increasing. Since the upper bound the $\tau_\delta$ is the minimum $u$ such that $h(u) > c_1 n$ (according to Equ. (11)). Therefore, we have $\tau_\delta \leq \lceil \tau' \rceil$ where $h(\tau') = c_1 n$ and $\tau' > e^{d+1}$. Consequently, we have:

$$
\begin{aligned}
\ln(\tau')^{d+1} &= \tau'/c_1 n \\
\ln(\tau') &= \tau'^{1/(d+1)}/(c_1 n)^{1/(d+1)} .
\end{aligned}
$$

Denote $\tilde{\tau} := \tau'^{1/(d+1)}$ (monotonically increasing w.r.t. $\tau'$). We have:

$$
\begin{aligned}
\ln(\tilde{\tau}^{(d+1)}) &= \tilde{\tau}/(c_1 n)^{1/(d+1)} \\
(d+1)\ln(\tilde{\tau}) &= \tilde{\tau}/(c_1 n)^{1/(d+1)} \\
\ln(\tilde{\tau}) &= \tilde{\tau}/\big((d+1)(c_1 n)^{1/(d+1)}\big) \\
\tilde{\tau} &= e^{\frac{\tilde{\tau}}{(d+1)(c_1 n)^{1/(d+1)}}}
\end{aligned}
$$

Here, we introduce Lambert $W$ function $x = W(y)$, s.t. $y = xe^x$. Let $y = -1/c$ and $x = -\frac{x'}{c}$, we have that $x' = -cW(-1/c)$, s.t. $1 = \frac{-x'}{e^{x'/c}}$. Therefore, we have that $\tilde{\tau} = -c_2 W(-1/c_2)$ where $c_2 = (d+1)(c_1 n)^{1/(d+1)}$. Since $-1/2 \leq -1/c_2 \leq 0$ and according to (Chatzigeorgiou, 2013), $W$ has two real-valued branch $W_0$ and $W_{-1}$ and $W_0(x) \geq -1$ in this case. Therefore, if $W = W_0$, we have $\tilde{\tau} \leq -c_2 * (-1) = c_2$. In this case $\tilde{\tau} = \mathcal{O}(n^{1/(d+1)})$. Therefore, $\tau' = \tilde{\tau}^{d+1} = \mathcal{O}(n)$. Consequently $\tau_\delta = \mathcal{O}(n)$. If $W = W_{-1}$, according to Lemma C.8 where we let $x = \ln(c_2) - 1$, we have:

$$
W(-1/c_2) > -1 - \sqrt{2(\ln(c_2) - 1)} - (\ln(c_2) - 1) .
$$

Hence

$$
\tilde{\tau} = -c_2 W(-1/c_2) < c_2 + c_2\sqrt{2(\ln(c_2) - 1)} + c_2(\ln(c_2) - 1) .
$$

Therefore, we get $\tilde{\tau} = \mathcal{O}(c_2 \ln(c_2)) = \mathcal{O}(n^{1/(d+1)} \log(n))$. Consequently, $\tau' = \tilde{\tau}^{d+1} = \mathcal{O}(n \log^{d+1}(n))$. Hence we have $\tau_\delta = \mathcal{O}(n \log^{d+1}(n))$. Following the same proof technique, we have that when $k$ is a linear kernel, $\tau_\delta = \mathcal{O}(n \log n)$. $\qquad \square$

## C.4 PROOF FOR PROPOSITION 4.5

We need to proof the following result first:

**Lemma C.9.** For an ranked values $\varphi_1 \geq \varphi_2 \geq \varphi_m > \varphi_{m+1} \geq \cdots \geq \varphi_n$, each element $\varphi_i$ is added with a number $b_i$ which satisfies $|b_i| \leq \varepsilon'$ and hence results in $\varphi_i' = \varphi_i + b_i$. Rank the new values $\{\varphi_i'\}_{i \in N}$ from high to low and denote the $m$-th ranked value as $\varphi_{o_m}'$. We have that $|\varphi_{o_m}' - \varphi_m| \leq \varepsilon'$.

*Proof.* Denote the $\mathcal{I}_m^* = \{1, \ldots, m\}$ and denote $\mathcal{I}_m'$ the indices of the top-$m$ values among $\{\varphi_i'\}_{i \in N}$ where $|\mathcal{I}_m'| = m$. Note that $\mathcal{I}_m'$ might not be unique. Denote $i' = \arg\min_{i \in \mathcal{I}_m'} \varphi_i'$. We consider the following situations

(a) If $i' \leq m$, we have that $\varphi_{i'} \geq \varphi_m$. According to the definition of $i'$, we have that $\varphi_{o_m}' = \varphi_{i'} + b_{i'} \leq \varphi_m + b_m$. Hence $\varphi_{o_m}' - \varphi_m \leq b_m$. We have that $\varphi_{o_m}' = \varphi_{i'} + b_{i'} \geq \varphi_m + b_{i'}$. Hence, we have $\varphi_{o_m}' - \varphi_m \geq b_{i'}$. Consequently, we have that $|\varphi_{o_m}' - \varphi_m| \leq \max(|b_m|, |b_{i'}|) \leq \varepsilon'$.

(b) If $i' > m$, there exits $j \in \mathcal{I}_m^*$ such that $\varphi_j + b_j \leq \varphi_{o_m}'$. This is because intuitively $i' \in N \setminus \mathcal{I}_m^*$ is included in $\mathcal{I}_m'$ and hence there exists at least an element in $\mathcal{I}_m^*$ will be excluded from $\mathcal{I}_m'$ and to be included in $N \setminus \mathcal{I}_m'$ such that $|\mathcal{I}_m'| = m$. Hence $\varphi_{o_m}' - \varphi_j \geq b_j$. Since we have that $\varphi_{i'} < \varphi_m$ according to the definition, $\varphi_{o_m}' = \varphi_{i'} + b_{i'} < \varphi_m + b_{i'}$. Consequently, we have $\varphi_{o_m}' - \varphi_m < b_{i'}$. Finally, we get $|\varphi_{o_m}' - \varphi_m| \leq \max(|b_j|, |b_{i'}|) \leq \varepsilon'$.

Therefore, in all cases, $|\varphi_{o_m}' - \varphi_m| \leq \varepsilon'$. $\qquad\square$

*Proof of Proposition 4.5.* Denote $\varphi_{o_m}(p)$ the $m$-th largest value among $\{\varphi_i(p)\}_{i \in N}$. Define the following value:

$$\varphi_i'(p) = \frac{1}{n} \sum_{k=1}^{p} \Delta_i^k .$$

Denote $\mathcal{I}$ an $(\varepsilon, \delta)$-approximation to the top-$m$ of $\{\varphi_i(p)\}_{i \in N}$ where $\mathcal{I}$ is the index set with $|\mathcal{I}| = m$. Hence we have that with probability at least $1 - \delta$

$$\varphi_j(p) \geq \varphi_{o_m}(p) - \varepsilon, \forall j \in \mathcal{I} .$$

Since $\varphi_i(p) = \frac{p}{n}\varphi_i'(p)$, we have that $\frac{p}{n}\varphi_j(p) \geq \frac{p}{n}\varphi_{o_m} - \frac{p}{n}\varepsilon$, which gives

$$\varphi_j'(p) \geq \varphi_{o_m}'(p) - \frac{p}{n}\varepsilon, \forall j \in \mathcal{I} .$$

Since $|\Delta_i^k| \leq \varepsilon', \forall k \in \{p, \ldots, n\}$ according to the assumption, we have

$$\begin{aligned}
|\varphi_i - \varphi_i'(p)| &= |\frac{1}{n} \sum_{k=p+1}^{n} \Delta_i^k| \\
&\leq \frac{1}{n} \sum_{k=p+1}^{n} |\Delta_i^k| \\
&\leq \frac{n-p}{n}\varepsilon' .
\end{aligned}$$

According to Lemma C.9 we have

$$|\varphi_{o_m}'(p) - \varphi_m| \leq \frac{n-p}{n}\varepsilon'$$

Hence, we get the following

$$\varphi_j + \frac{n-p}{n}\varepsilon' \geq \varphi_j'(p) \geq \varphi_{o_m}'(p) - \frac{p}{n}\varepsilon \geq \varphi_m - \frac{n-p}{n}\varepsilon' - \frac{p}{n}\varepsilon, \forall j \in \mathcal{I} .$$

Hence, with probability at least $1 - \delta$

$$\varphi_j \geq \varphi_m - \frac{2(n-p)}{n}\varepsilon' - \frac{p}{n}\varepsilon, \forall j \in \mathcal{I} .$$

In other words, $\mathcal{I}$ is an $(\frac{2(n-p)}{n}\varepsilon' + \frac{p}{n}\varepsilon, \delta)$-approximation to the top-$m$ of $\{\varphi_j\}_{j \in N}$. Equivalently, we have that an $(\frac{n}{p}\varepsilon - \frac{2(n-p)}{p}\varepsilon', \delta)$-approximation to the top-$m$ of $\varphi_i(p)_{i \in N}$ is also an $(\varepsilon, \delta)$-approximation of the top-$m$ of $\{\varphi_i\}_{i \in N}$. $\qquad\square$

## C.5 Proof for Equ. (8)

*Proof.* We restate the efficiency, symmetry, and additivity axioms of SV from (Shapley et al., 1953; Roth, 1988):

- **Efficiency.** The sum of SV equals the utility of the grand coalition, i.e., $\sum_{i \in N} \varphi_i = U(N)$.

- **Symmetry.** If $U(S \cup \{i\}) = U(S \cup \{j\}), \forall S \in N \setminus \{i, j\}$, we have $\varphi_i = \varphi_j$.

- **Null player.** If $U(S \cup \{i\}) = U(S), \forall S \in N \setminus \{i\}$, we have $\varphi_i = 0$.

- **Additivity.** Denote $\varphi_i^{(j)}$ SV defined by $U_j$. If $U(S) = \sum_j U_j(S), \forall S \subseteq N$, we have $\varphi_i = \sum_j \varphi_i^{(j)}$.

Since our utility function defined in Equ. (7) can be rewritten as:

$$U(S) = \sum_{z_j \in D_V} U_j(S) ,$$
$$U_j(S) := \frac{1}{|D_V|} \mathbb{1}\big(g(z_i) \in M(D_S, \varepsilon)\big) . \tag{12}$$

Therefore, according to the additivity of SV, we have $\varphi_i = \sum_j \varphi_i^{(j)}$ where $\varphi_i^{(j)}$ is SV defined by $U_j(S)$ in Equ. (12).

Denote $D_V' = \{z_j \in D_V | \exists z_k \in D_N, \rho(g(z_k), g(z_j)) \leq \varepsilon\}$, i.e., data points in the validation dataset that have at leat one data point from the training dataset $D_N$ that is within $\varepsilon$ distance to them. In this case we have $U_j(N) = \frac{1}{|D_V|}, \forall z_j \in D_V'$. Denote the set $D_j' = \{z_k \in D_N : \rho(g(z_k), g(z_j)) \leq \varepsilon\}$ where $z_j \in D_V'$. In this case since every $z_k \in D_V, z_k \notin D_j'$ we have $U_j(S \cup \{k\}) = U_j(S)$. According to null player,

$$\varphi_k^{(j)} = 0, \forall z_k \in D_V, z_k \notin D_j' . \tag{13}$$

While for $z_k, z_l \in D_j'$, we have that $U(S \cup \{k\}) = U(S \cup \{l\}), \forall S \subseteq N\{k, l\}$, and hence $\varphi_k^{(j)} = \varphi_l^{(j)}$ for all $z_k, z_l \in D_j'$ (i.e., symmetry). We have that $\sum_{k \in N} \varphi_k^{(j)} = U_j(N) = \frac{1}{|D_V|}$ (i.e., efficienty). Therefore we have:

$$\varphi_k^{(j)} = \frac{1}{|D_j'||D_V|}, \forall z_k \in D_j' \tag{14}$$

Combining Equ. (13) and Equ. (14), we have:

$$\varphi_k^{(j)} = \frac{\mathbb{1}\big(\rho(g(z_k), g(z_j)) \leq \varepsilon\big)}{|D_j'||D_V|}, \forall z_j \in D_V' . \tag{15}$$

Similarly, for $z_j \notin D_V', z_j \in D_V$, we have that $U_j(N) = 0$ and according to the efficiency and symmetry, we have that $\varphi_k^{(j)} = 0, \forall k \in N$. According to the additivity, we have that:

$$\varphi_k = \sum_{z_j \in D_V} \varphi_k^{(j)} = \sum_{z_j \in D_V'} \varphi_k^{(j)} = \sum_{z_j \in D_V'} \frac{\mathbb{1}\big(\rho(g(z_k), g(z_j)) \leq \varepsilon\big)}{|D_j'||D_V|}, \forall k \in N .$$

$\square$

## C.6 Proof for Proposition A.2

*Proof.* **Efficiency.** According to the definition in Equ. (9), we have

$$\sum_{i \in N} \varphi_i^{(m)} = \sum_{i \in \mathcal{S}_m^*} U(N)/m = U(N) .$$

Therefore efficiency holds.

**Symmetry.** According to the symmetry of SV, if $U(S \cup \{i\}) = U(S \cup \{j\}), \forall S \subseteq N \setminus \{i, j\}$, we have $\varphi_i = \varphi_j$. According to the assumption that $\varphi_1 \geq \varphi_2 \geq \cdots \geq \varphi_m > \varphi_{m+1} \geq \varphi_{m+2} \geq \cdots \geq \varphi_n$, e have that either $i, j \in \mathcal{S}_m^*$ or $i, j \notin \mathcal{S}_m^*$, in either case, we have $\varphi_i^{(m)} = \varphi_j^{(m)}$ according to the definition in Equ. (9).

**Strict $m$-th desirability.** According to the strict desirability of SV (Maschler & Peleg, 1966), we have that $(\exists B \in N \setminus \{i, j\}, U(B \cup \{i\}) > U(B \cup \{j\})) \wedge (\forall C \in N \setminus \{i, j\}, U(B \cup \{i\}) \geq U(B \cup \{j\})) \implies \varphi_i > \varphi_j$. Therefore, we have that $(\exists B \in N \setminus \{i, j\}, U(B \cup \{i\}) > U(B \cup \{j\})) \wedge (\forall C \in N \setminus \{i, j\}, U(B \cup \{i\}) \geq U(B \cup \{j\})) \wedge (\varphi_j = \varphi_{m+1}) \implies \varphi_i > \varphi_{m+1}$. According to the assumption that $\varphi_1 \geq \varphi_2 \geq \cdots \geq \varphi_m > \varphi_{m+1} \geq \varphi_{m+2} \geq \cdots \geq \varphi_n$, we have $i \in \mathcal{S}_m^*$ and $j \notin \mathcal{S}_m^*$ (since $\varphi_j = \varphi_{m+1}$). Therefore, $\varphi_i^{(m)} = \frac{U(N)}{m} > 0$ (since $U(N) > 0$ according to the assumption) and $\varphi_j^{(m)} = 0$ according to the definition in Equ. (9). Hence $\varphi_i^{(m)} > \varphi_j^{(m)}$. $\qquad\square$

