# OpenReview forum: "Efficient Top-m Data Values Identification for Data Selection"
_ICLR.cc/2025/Conference — ICLR 2025 Poster_

### Official Review · Reviewer_PSf4 · 2024-11-03

**Soundness:** 3
**Presentation:** 3
**Contribution:** 3
**Rating:** 8
**Confidence:** 3

**Summary:**

The paper proposes a bandit-based method for identifying the top-data from a dataset in terms of data values. The main contribution is the incorporation of data features, specifically a non-linear structure assumed via a Gaussian process. The proposed algorithm is shown to have a PAC guarantees and is validated through a set of experiments on different applications.

**Strengths:**

It is an interesting combination of Gaussian processes and data valuation problem. The proposed solution addresses the problem where the linearity assumption is impractical for some applications, which is important. The algorithm is analyzed within the common bandit framework for its correctness and stopping time. A good amount of comprehensive experiments are included to demonstrate the effectiveness.

**Weaknesses:**

- It is a bit misleading in Theorem 4.2 and Proposition 4.3 that the same notation $\tau_\delta$ is called stopping iteration and query complexity differently.
- The query complexity / stopping time is lack of comparison with any prior works or lower bounds, making it a bit hard to evaluate the theoretical contribution.

**Questions:**

Minor issue: It seems not all of the experiments include error bars? What is the confidence level?

---

> ### Author Response · Authors · 2024-11-21
>
> We would like to thank the reviewer for taking the time to review our paper. We also thank the reviewer for acknowledging that the problem we are studying is important, our proposed algorithm is interesting and our experiments are adequate. We would like to address your specific questions below.
>
> > It is a bit misleading in Theorem 4.2 and Proposition 4.3 that the same notation $\tau_\delta$ is called stopping iteration and query complexity differently.
>
> Thanks for pointing this out. Note that since we only make one query in every iteration, the stopping iteration equals the number of queries made in our algorithm. We have made this clearer by adding this discussion in our updated PDF.
>
> > The query complexity / stopping time is lack of comparison with any prior works or lower bounds, making it a bit hard to evaluate the theoretical contribution.
>
> Indeed our work is the first to study the identification of top-$m$ data values. There are no existing works that study the complexity of the baseline methods, rendering our method difficult to compare with existing baselines. However, existing works do give a query complexity of $O(n^2\log n)$ for Monte-Carlo-based methods to approximate the Shapley values directly (which does not translate to top-$m$ data values identification directly due to different metrics). This can be a peek for the improvement of our method over Monte-Carlo-based methods. The detailed analysis of existing works' query complexity in terms of top-$m$ data values identification can be itself a major contribution and hence in future work.
>
>
>
> > Minor issue: It seems not all of the experiments include error bars? What is the confidence level?
>
> We wish to point out most of our experiments have error bars except for Fig. 2, 4, and 5 due to the large scale of the problem (i.e., $n=10k$) and the expensive computation of the baseline methods. Specifically, it will be expensive to run the experiments on this problem scale multiple times for all baseline methods. However, each subplot in Fig. 2 shows the results for different settings (i.e., different $m$). Our algorithm outperforms other methods consistently among all these settings, which serves as a surrogate for the expensive error bars. We will add this discussion in our next revision.
>
> ---
>
> Thank you again for your time and your careful feedback. We hope our clarifications could improve your opinion of our work.

---

> > ### Comment · Reviewer_PSf4 · 2024-11-23
> >
> > Thank you for the response. I have increased my rating.

---

> > > ### Author Response · Authors · 2024-11-24
> > > **Thank you for raising the score**
> > >
> > > Thank you so much for your positive feedback! We are happy that we have addressed your concerns.

---

### Official Review · Reviewer_kM2u · 2024-11-04

**Soundness:** 3
**Presentation:** 3
**Contribution:** 3
**Rating:** 6
**Confidence:** 3

**Summary:**

This paper studies top-m Shapley value identification problem, formulated as top-m arm identification in multi-armed bandit. While traditional multi-armed bandit problems estimate arm values use reward feedback only, this paper utilizes data features and uses the Gaussian process to model the nonlinear mapping from data features to data values. The authors prove theoretically that the Gaussian process based top-m nonlinear bandit algorithm finds an $(\epsilon, \delta)$ approximation to the top-m data values. Extensive numerical experiments on three different tasks are provided to demonstrate the model performance.

**Strengths:**

This paper improves on existing literature by removing the linearity constraint of arm feature and arm value mapping. Different aspects (correctness, stopping iteration and query complexity) of the theoretical analysis are carefully addressed and compared to the linear baseline. Computational costs are further reduced by querying marginal contributions only on small subsets. Extensive experiments are conduct to convince the reader that the method indeed outperforms existing methods.

**Weaknesses:**

Gaussian process is commonly known as computationally expensive when it comes to online updating. At each iteration t, new data points are decided to query and then GP is updated with the improved candidate set $J(t)$. It would be good to discuss what methods or tricks can be used to decrease the computational cost for this setting.

**Questions:**

• What is the benefit of using GP instead of directly using neural networks or other dimension reduction methods?
    • What is the rate of diminishing return in Section 4.2 – is p here assumed to be of scale Cn (C is a constant)?

——————————After Rebuttal—————————Thank you for your response. I’m surprised to see the GP significant improvement without the newest tricks. I don’t have further questions and I’ll keep my current score.

---

> ### Author Response · Authors · 2024-11-21
>
> We would like to thank the reviewer for taking the time to review our paper. We would like to address your specific questions below.
>
> > Gaussian process is commonly known as computationally expensive when it comes to online updating. At each iteration t, new data points are decided to query and then GP is updated with the improved candidate set $J(t)$. It would be good to discuss what methods or tricks can be used to decrease the computational cost for this setting.
>
> Indeed, online updating GP requires some amount of computation. There are two main ways from previous GP works that can be used to reduce this computational cost.
>
> 1. Instead of updating GP in every iteration, we can update GP every few iterations (i.e., the batch setting), which has been used in our experiments in Fig. 2. Specifically, we only update our GP every 100 queries and every iteration we will select the top 100 data points that maximize the gap indices (i.e. Line 5-7 of Algorithm 1) to make queries. This reduces the computational cost by a lot, and our GPGapE still outperforms other approaches by a large margin (as shown in Fig. 2).
>
>
> 2. We can also reduce the computational complexity of GP by using the previous work on sparse Gaussian processes [1]. Specifically, the sparse Gaussian process uses a subset of inducing points to approximate the full Gaussian process, hence removing the requirement of running GP on the full dataset. Therefore, this can be a plugin to our current algorithm to immediately improve computational efficiency.
>
> Thank you for pointing this out. We will add this discussion in our next revision.
>
>
> [1] Quinonero-Candela, J., & Rasmussen, C. E. (2005). A unifying view of sparse approximate Gaussian process regression. The Journal of Machine Learning Research, 6, 1939-1959.
>
> > What is the benefit of using GP instead of directly using neural networks or other dimension reduction methods?
>
> The main reason for choosing GP over neural network is that GP allows us to derive the upper confidence bound (i.e., Equ. (4)) for the true gap of data values (i.e., defined in Line 213) and this upper confidence bound is used as an essential part of our algorithm design (i.e., Line 5-6 of Algorithm 1) and hence allows us to derive the theoretical results for our algorithm (i.e., Theorem 4.1&4.2 and Proposition 4.3). However, it is unclear how to derive this upper confidence bound for neural networks hence making it inapplicable in our algorithm. It is possible to use the existing works on Bayesian neural networks [2] for deriving this upper confidence bound. However, how to derive this upper confidence bound in that case and how it affects our theoretical results are unclear and can be studied in future work.
>
> For dimension reduction methods, note that they are orthogonal to GP in this case, meaning that these methods can be a plugin for our GP to make GP more efficient. In fact, we have used PCA in our experiments to reduce the dimension of data features to make our GP more effective (in Lines 716-718).
>
> [2] Goan, E., & Fookes, C. (2020). Bayesian neural networks: An introduction and survey. Case Studies in Applied Bayesian Data Science: CIRM Jean-Morlet Chair, Fall 2018, 45-87.
>
> > What is the rate of diminishing return in Section 4.2 – is p here assumed to be of scale Cn (C is a constant)?
>
> The rate of diminishing return depends on the size of the dataset $n$ and also the difficulty of the tasks hence not a fixed rate. Specifically, for a very large dataset (e.g., $n=100k$), $p$ can be very small compared to the size of $n$ since the marginal contribution of a data point will be close to $0$ when $p$ is much less than $n$. For tasks that are difficult (e.g., image data tasks require much more data to learn a good model and hence more difficult compared to tabular data tasks with a few features), they generally require more data to achieve a good performance, in that case, $p$ can be very close to $n$ since adding more data may still be helpful in improving the model performance.
>
> Empirically, we set $p$ to be $0.3n$ which already reduces the wall time of our algorithm a lot as we have shown in Table 2 without sacrificing the performance.
>
> ---
>
> Thank you again for your time and your careful feedback. We hope our clarifications could improve your opinion of our work.

---

> ### Author Response · Authors · 2024-11-25
> **We would like to know if you have any further questions that require additional clarifications.**
>
> Dear Reviewer kM2u,
>
> Thank you again for your time in reviewing our paper and for asking many valuable questions.
>
> Please let us know if our replies have addressed your concerns. We would be happy to address any further questions you might have within the discussion period.
>
> Best regards,
>
> Authors

---

### Official Review · Reviewer_gPqR · 2024-11-04

**Soundness:** 3
**Presentation:** 3
**Contribution:** 3
**Rating:** 8
**Confidence:** 3

**Summary:**

This paper studies the problem of identifying the top-m data points with the largest Shapley Value (SV), which can be viewed as a top-m best arm identification problem. The paper models the relationship between the data feature and the value as a Gaussian Process (GP) and proposes an algorithm called GPGapE. The algorithm uses GP to update the estimation statistics and then designs which arm to sample based on GapE, a popular top-m best arm identification algorithm. Theoretical analysis of the sample complexity and PAC guarantee is provided, and extensive numerical experiments are conducted to showcase the performance of GPGapE. Interestingly, the paper makes use of the diminishing returns of ML models to propose a simplification of GPGapE, which achieves even better empirical performance.

**Strengths:**

1. Formulation and Algorithm: This paper combines the GP with the top-m best arm identification. The proposed GPGapE algorithm is novel, and the studied top data values identification problem has practical value.

2. Analysis: The sample complexity and the PAC guarantee are analyzed theoretically under the standard assumptions of both GP and top-m best arm identification. Extensive numerical experiments are provided to showcase the performance of the proposed algorithm

3. Computation Efficiency: A simpler version of GPGapE is also studied to reduce the computational complexity, where the paper makes equivalence arguments that the performance of GPGapE can be achieved using the simpler version. The numerical experiments interestingly show that the simpler version achieves even better performance.

**Weaknesses:**

1. GP: This paper inherits the weaknesses of the GP literature, which requires strong assumptions that the studied function is in RKHS. And all theoretical results are based on it. It remains a little unclear to me for general data value modeling, why GP is a good approximation that can provide the significant improvement shown in the numerical section. Justifications should be more detailed.

2. Sub-Gaussian: The authors also mentioned in the paper that some utility functions such as cross-entropy loss are not bounded, justifications on why sub-Gaussian noise is still a good assumption in this setting should be given and detailed.

3. Presentation: The presentation of the paper could be improved, notations are used without illustration and are causing ambiguity, e.g., in Theorem 4.1, when choosing C_{\delta,t}, what is B? the format of information gain \gamma_t is not explained, and how do you know the random stopping time \tau_\delta ahead of time to set the parameter \lambda in GP?

**Questions:**

1. For Theorem 4.1, Theorem 4.2, and Proposition 4.3, can we choose the same hyperparameter inputs to obtain all results simultaneously? Can we preset the inputs ahead of time or does it require tuning?

2. The current algorithm requires storing all historical data feature-marginal contribution pairs. Could you provide some insights on whether this is theoretically necessary? or is there a more memory-efficient implementation?

---

> ### Author Response · Authors · 2024-11-21
> **Author Response Part 1/2**
>
> We would like to thank the reviewer for taking the time to review our paper. We also thank the reviewer for acknowledging that the problem we are studying is practical, our proposed algorithm is novel and our experiments are extensive. We would like to address your specific questions below.
>
>
> > GP: This paper inherits the weaknesses of the GP literature, which requires strong assumptions that the studied function is in RKHS. And all theoretical results are based on it. It remains a little unclear to me for general data value modeling, why GP is a good approximation that can provide the significant improvement shown in the numerical section. Justifications should be more detailed.
>
> We wish to point out that we did provide detailed both theoretical and empirical justifications on why GP is a good modeling choice for data values in Appendix A.4, titled "On the rationale of choosing GP to model the data value mapping function". We will repeat it here for easier reference:
>
> **Theoretical justification:** The difference of Shapley value of $i$ and $j$ can be bounded by the distance of the data point $i$ and data point $j$ according to the following:
>
> Lemma (Similarity-bounded difference in the Shapley value) [Xu et al. 2024, Lemma 6]. For all $i,j\in N$, $\bigl(\forall S \subseteq N\setminus\{i,j\})\quad |U(S\cup\{i\})-U(S\cup\{j\})| \le Ld(i,j)\bigl) \implies |\psi_i -\psi_j| \le ZLd(i,j)$ where $L \ge 0$ is a constant and $d(i,j)$ is some distance measure between $i$ and $j$, and $Z$ is the linear scaling parameter.
>
> The condition for the lemma to hold is that if two data points $i$ and $j$ are similar in data space, the performance of dataset $S\cup\{i\}$ and $S\cup\{j\}$ should be similar. This should be trivially true since these two data points contribute similar information to the model (e.g., data from a specific subgroup in the data space) and hence the model performance should be similar.
>
> **Empirical justification:** To give an empirical verification, we compute the distance of Shapley value and the distance of data feature for randomly sampled data point pairs from the dataset. Fig. 8 in our original paper shows that when the distance of the data feature increases, the distance of Shapley values also increases, which validates the theoretical result above.
>
>
> **Since similar data points have similar Shapley values, GP is a good design choice to model the mapping function.** Specifically, GP essentially uses the kernel-based method to do prediction, and the idea of kernel function is based on the belief that similar input should have similar function output. Consequently, GP is a good choice to model the mapping function.
>
> > Sub-Gaussian: The authors also mentioned in the paper that some utility functions such as cross-entropy loss are not bounded, justifications on why sub-Gaussian noise is still a good assumption in this setting should be given and detailed.
>
> Indeed, some of the utility functions $U(S)$ are not bounded (e.g., cross-entropy loss). However, since we are looking at the noise of the marginal contribution, i.e., $U(S\cup\{i\}) - U(S)$, it can still be bounded even if $U(S)$ is not bounded. Specifically, the marginal contribution, in this case, is the change of cross entropy loss when a single data point is added to the data subset $S$ and empirically this change will be very small (i.e., usually at the magnitude of less than 1e-2), especially when the size of $S$ is moderately large due to the diminishing return as we described in Fig. 1. Therefore, we are still able to bound the marginal contribution empirically and hence the sub-Gaussian condition is easily satisfied.

---

> ### Author Response · Authors · 2024-11-21
> **Author Response Part 2/2**
>
> > Presentation: The presentation of the paper could be improved, notations are used without illustration and are causing ambiguity,
> > in Theorem 4.1, when choosing C_{\delta,t}, what is B?
>
> $B$ is the upper bound of the RKHS norm of the function $f$ as described in Line 255.
>
> > the format of information gain \gamma_t is not explained,
>
> Thanks for pointing out. The maximum information gain is defined in Srinivas 2010, and to introduce this will need an extra amount of notations in our paper. We will add a detailed definition in our appendix for easier reference.
>
> > and how do you know the random stopping time \tau_\delta ahead of time to set the parameter \lambda in GP?
>
>
> In fact, for all our theorem to hold, we only require $\lambda$ to be set, such that $1 < \lambda \le 1+2/\tau_\delta$, i.e., a value that is very close to 1. The same technique has been used in Theorem 2 of Chowdhury & Gopalan, 2017.
>
> On the other hand, the selection of $\lambda$ here is just for the theoretical result. In practice, we can choose $\lambda$ to be a fixed value based on the level of noise in the observation (i.e., the variance of $\eta_t$ in Line 105) and the algorithm turns out to perform well and achieves the SOTA performance in our experiments in Sec. 5.
>
> Thanks for pointing this out, we will make all these points clearer in our next revision.
>
> > For Theorem 4.1, Theorem 4.2, and Proposition 4.3, can we choose the same hyperparameter inputs to obtain all results simultaneously? Can we preset the inputs ahead of time or does it require tuning?
>
> Theoretically, Theorem 4.1, Theorem 4.2, and Proposition 4.3 can hold at the same time with the same hyperparameter. Specifically, Theorem 4.1, and Theorem 4.2 are dependent on the same group of assumptions and hence hold with the same hyperparameters. Proposition 4.3 will require an addition condition of the kernel function to be less than 1 (trivially true for both kernel functions that we are considering) and gives different results based on different kernel functions selected in Theorem 4.1, Theorem 4.2. Hence, these three theoretical results can hold with a single group of hyperparameters.
>
> Empirically, we will need to determine which kernel function will suit a certain task well. In all our experiments we use RBF and outperform other baselines already which shows the robustness of our algorithm. Therefore, we can use RBF ahead of time which gives the best results already. Alternatively, a more careful tuning of kernel functions on specific tasks can improve the empirical results.
>
> > The current algorithm requires storing all historical data feature-marginal contribution pairs. Could you provide some insights on whether this is theoretically necessary? or is there a more memory-efficient implementation?
>
> Indeed, the exact Gaussian process requires storing all the historical data. However, this can be improved by using the previous work on sparse Gaussian processes [1]. Specifically, the sparse Gaussian process uses a subset of inducing points to approximate the full Gaussian process, hence removing the requirement of storing all historical data since only this subset of inducing points is required. Therefore, this can be a plugin to our current algorithm to immediately improve memory efficiency. However, reducing the memory requirement for GP is not the focus of our work. We will add this discussion in our next revision.
>
> [1] Quinonero-Candela, J., & Rasmussen, C. E. (2005). A unifying view of sparse approximate Gaussian process regression. The Journal of Machine Learning Research, 6, 1939-1959.
>
> ---
>
> Thank you again for your time and your careful feedback. We hope our clarifications could improve your opinion of our work.

---

> > ### Comment · Reviewer_gPqR · 2024-11-22
> >
> > I appreciate the detailed response from the authors, which indeed addresses most of my concerns, especially the motivation for GP. Overall I believe this paper studies a relevant problem where the use of GP and top-m best arm identification are natural. Both the theory and experiments look solid with one minor issue: setting the $\lambda$ parameter of the GP to achieve theoretical guarantees requires knowledge of an upper bound of the sample complexity $\tau_\delta$. I appreciate the idea of using diminishing returns to simplify GPGapE and am surprised to see its performance.
> >
> > Based on my judgment, I'm happy to increase my score from 6 to 8. However, since I'm not an expert in the field of data value identification, I maintain moderate confidence 3.

---

> > > ### Author Response · Authors · 2024-11-22
> > > **Thank you for acknowledging our response and raising the score**
> > >
> > > Thank you so much for your positive feedback! We are happy that we have addressed your concerns. We will add more discussion on the selection of $\lambda$ in the revised version of our paper.

---

### Official Review · Reviewer_7tcQ · 2024-11-07

**Soundness:** 3
**Presentation:** 3
**Contribution:** 2
**Rating:** 5
**Confidence:** 3

**Summary:**

The authors consider the problem of identifying a small set of samples in a data set with high value for training.  Shapley value is used for measuring individual samples’ value, but is prohibitive to compute.  The authors propose that since only identifying the samples is necessary, not identifying their values, this problem can be viewed as a top-m multi-armed bandits problem to more efficiently solve it.  Furthermore, since training samples’ values may depend on rich features, similarity in (non-linear) feature space can be modeled using Gaussian processes.  The authors propose a top-m GP algorithm, including with a variant involving sampling rewards via computing Shapley values using small cardinality sets.  The authors also analyze regret and provide experiments.

**Strengths:**

- The motivating problem of identifying valuable samples in large data sets is challenging but important
- The problem of fixed confidence top-k Gaussian process bandits is interesting
- The presentation overall is good – I found the paper to be well organized and written
- The authors identified high-probability guarantees of their method and show for linear and RBF kernel functions nearly linear query complexity (in terms of # of samples)
- The use of GPs (compared to linear) for feature mappings is good.
- The authors include a number of experiments

**Weaknesses:**

### Major
- I think the technical problem setting (top-m bandit) is poorly motivated for the domain problem (Shapley value). If you are only going to select $m\ll N$ data points, why would you use Shapley value as the utility?  Shapley value is a fair allocation of joint/group value, which could be very different than what is achieved by groups of size $m$.  If it were easy to compute, I could see it potentially as a heuristic, but given the challenge in computing it, in some sense is it worth putting effort into using as a metric when it does not necessarily tell us how different groups of size $m$ will perform?
    - for top $m$, wouldn’t we either care about the top m data points in terms of marginal value to the primary utility function $U$ over which Shapley value is to be computed, eg if $U$ (1) is the overall utility function then $U({x_i})$ if the data points could be used/sold individually, or their value as a set A of size $m$, $U(A)$.  It seems implicit in your set up that you are looking at the top $m$ in terms of an alternate utility function, namely the Shapley value, which depends on both the actual utility of interest $U$ and set of all data $N$, looking at arms whose individual Shapley values are the highest.

### Minor
- Lines 313-320 there is an argument that since individual data points tend to have diminishing value with larger data sets, one can sample smaller subsets S to get bounds; certainly if (a) that holds uniformly and (b) the computational complexity increases with cardinality of $S$, that seems smart.  But is that diminishing returns empirically true for most samples or for samples on average?  Eg in Figure 1 is that for randomly sampled points?  Did you also check if that holds for the top $m$ points too, esp if $m\ll N$?  For instance, I don’t work in vision or NLP, but I could imagine that for some samples of rare classes, if there’s too few samples the ML model would not learn about those rare classes and those samples could have low Shapley value (not contributing to the learned common classes), but with larger $N$, those rare classes could be learned by the ML model leading to higher accuracy and subsequently higher Shapley value for those samples from rare classes.

- Proposition 4.4, Figure 1, and discussion between
    - Prop 4.4 statement includes an assumption $|\Delta_i^\ell| \leq \epsilon’, \ \forall \ell \in \{p,\dots,n\}$; it would be better to have that assumption (as in a theorem environment as Assumption 4.4) and discussed in relation to lines 313-323.
    - Figure 1 caption says “Diminish return of adding new data points to the dataset when the size of the dataset increases.” But the x-axes are $l$, the subset size $S$.  “adding new data points to the dataset” means going from $N$ data points to $N+10$ or $N+100$.  In the main text, it mentions both “adding new data” (line 314) and (lines 319-320) that $|\Delta_i^l|$ is decreasing in $l$; both phenomena may happen and potentially be useful for speeding up methods.
    -  The Prop 4.4 statement (at least per sample $i$) $|\Delta_i^\ell| \leq \epsilon’, \ \forall \ell \in \{p,\dots,n\}$ seems to be combining both  “adding new data points to the dataset” and $|\Delta_i^l|$ is decreasing in $l$, as it is bounding $|\Delta_i^l|$ in an absolute sense seemingly implicit that for sufficiently large $N$, not only will $|\Delta_i^l|$ be small (small enough one could use the bound as a surrogate $\epsilon$ for the top-m threshold) but also that (N fixed) there will be a subset cardinality $l=p$ for which all larger cardinalities will not have a larger value.


     – also $\exists p \in N$  should there be a $\forall i\in N$ there?  The conclusion involves bounding all $\varphi_i$ for all $i\in N$



### Very minor (Typos/etc.)
- for subset $S$ cardinality, $\ell$ can be easy to read as super/sub script than $l$
- Figure 1 caption “Diminish return”
- Figure 7 – the curves are quite crowded; hard to tell, but it might be a little easier to read if the curves are transparent

**Questions:**

- Can you discuss what technical challenges there were for obtaining the high-probability guarantees and bounding query complexity, given prior work like Reda et al for the linear setting and past work on Gaussian process bandits?

---

> ### Author Response · Authors · 2024-11-21
> **Author Response Part 1/2**
>
> We would like to thank the reviewer for taking the time to review our paper. We also thank the reviewer for acknowledging that the problem we are studying is challenging but important, our proposed solution is interesting and our presentation is well organized. We would like to address your specific questions below.
>
>
> > If you are only going to select $m << N$ data points, why would you use Shapley value as the utility? Shapley value is a fair allocation of joint/group value, which could be very different than what is achieved by groups of size $m$... given the challenge in computing it, in some sense is it worth putting effort into using as a metric when it does not necessarily tell us how different groups of size $m$ will perform?
>
>
> We wish to point out that **Shapley value has been widely used for data selection in the existing works** [1,2,3,4], (Ghorbani & Zou, 2019; Nohyun et al., 2022). Specifically, these works have empirically shown the state-of-the-art data selection performance using the Shapley value among existing baselines. The work of [1] studies the optimality of data selection using the Shapley value and provides the following theoretical result:
>
> **Theorem 6 from [1].** For any utility functions $U$ that is monotonically transformed modular (defined in [1]), Shapley value is optimal for size-$k$ data selection tasks for any $k = 1, . . . , n − 1$.
>
> Your understanding of selecting a set of $A$ with size $m$ to directly maximize the $U(A)$ is correct, however, doing so is very expensive due to the combinatorial search space. Though the computation of Shapley value is expensive, it is still much cheaper than searching over the combinatorial space. Specifically, there are $\binom{n}{m}$ possible $A$ to evaluate their utility, i.e., $O(2^n)$ in terms of utility evaluation. Even if $m$ (e.g., $100$) is much smaller than $n$ (e.g., $100k$), $\binom{100k}{100}$ is still unbearable.
>
> On the other hand, **identifying top-$m$ data points with the largest Shapley values can be used not just in data selection but also in other applications.** It can also be used to identify noisy data points by looking at the lowest-$k$ Shapley values (empirically verified in Sec. 5.2). It can also be used to do data debugging (i.e., explainable ML). Specifically, when a trained model gives a certain prediction for a specific input $x$ we can use top-$m$ data values identification to find the top-$m$ data points that lead to this prediction in which Shapley value is proven to be effective in the work of [2].
>
> In summary, top-$m$ Shapley values identification has been shown to be an important problem due to its wide applications. Our work is the first to propose an efficient algorithm to identify the top-$m$ Shapley values compared to the previous works.
>
>
> [1] Wang, J. T., Yang, T., Zou, J., Kwon, Y., & Jia, R. (2024). Rethinking Data Shapley for Data Selection Tasks: Misleads and Merits. ICML 2024 Oral.
>
> [2] Wang, J., Lin, X., Qiao, R., Foo, C. S., & Low, B. K. H. (2024). Helpful or Harmful Data? Fine-tuning-free Shapley Attribution for Explaining Language Model Predictions. ICML 2024.
>
> [3] Tang, S., Ghorbani, A., Yamashita, R., Rehman, S., Dunnmon, J. A., Zou, J., & Rubin, D. L. (2021). Data valuation for medical imaging using Shapley value and application to a large-scale chest X-ray dataset. Scientific reports, 11(1), 8366.
>
> [4] Jiang, K., Liang, W., Zou, J. Y., & Kwon, Y. (2023). Opendataval: a unified benchmark for data valuation. Advances in Neural Information Processing Systems, 36.
>
>
>
> > ... But is that diminishing returns empirically true for most samples or for samples on average? Eg in Figure 1 is that for randomly sampled points? Did you also check if that holds for the top $m$ points too, esp if $m<<N$?
>
> The diminishing return is empirically true for almost all samples in terms of ML models. Specifically, Fig. 1 is for randomly selected points (as described in Lines 733-740) which may include top-$m$ data points. **To further check specifically whether this holds for top-$m$ data points we conduct additional experiments and add a plot for top-$m$ data points (both individual and averaged results) in Fig. 15 of our updated PDF.** The result is consistent with the Fig. 1 in the main paper.
>
> On the other hand, our theoretical result in Prop. 4.4 requires an even looser condition, i.e., the returns of a data point $i$ $|\Delta_i^l|$ be less than a certain value $\epsilon'$ when the size of the dataset $l$ is larger than a certain size. This does not require the return to be monotonically decreasing w.r.t. $l$ but just does not exceed a certain value after a certain size. Therefore, for each individual point (Fig. 15 left), even though $|\Delta_i^l|$ is not monotonically decreasing, the general trend is decreasing with minor ups and downs, which still supports our assumption of $\exists p \in N$ such that $|\Delta_i^l| \le \epsilon', \forall l\in \{p,\dots,n\}, \forall i \in N$ for Prop. 4.4.

---

> ### Author Response · Authors · 2024-11-21
> **Author Response Part 2/2**
>
> > For instance, I don’t work in vision or NLP, but I could imagine that for some samples of rare classes, if there’s too few samples the ML model would not learn about those rare classes and those samples could have low Shapley value (not contributing to the learned common classes), but with larger $N$, those rare classes could be learned by the ML model leading to higher accuracy and subsequently higher Shapley value for those samples from rare classes.
>
> We wish to point out that our claim is that $|\Delta_i^l|$ (i.e., not Shapley value) will decrease in general when $l$ (i.e., the size of the data subset that $i$ will be added to) increases. Note that the subsets with size $l$ are uniformly sampled from the full dataset. Specifically, in the case of data from a rare class, the data will improve the performance by more if it is added to a smaller subset since the smaller subset will have a lower chance of having data from this rare class hence adding this rare data will improve the model performance on this rare class by a lot. On the contrary, if the data subset size is large, the data subset will have a higher chance of having data from that rare class and hence adding this new rare data will improve the performance by less. Consequently, our claim still applies.
>
> > Prop 4.4 statement includes an assumption $|\Delta_i^\ell| \le \epsilon', \forall \ell \in p, \dots, n$; it would be better to have that assumption (as in a theorem environment as Assumption 4.4) and discussed in relation to lines 313-323.
>
> Thank you for pointing this out. Indeed it will be helpful to describe it in a separate assumption and discuss its relationship with the diminishing returns. We have separated this assumption and added this discussion in our updated PDF.
>
> > “Diminish return of adding new data points to the dataset when the size of the dataset increases.” But the x-axes are $\ell$, the subset size $S$, “adding new data points to the dataset” means going from $N$ data points to $N+10$ or $N+100$.
>
> By saying "when the size of the dataset increases.” we mean that the size of the data subset of the full dataset increases. Therefore it is correct to use $l$, i.e., the size of the data subset. By saying “adding new data points to the dataset” we mean adding data point $i$ to the data subset with size $l$. Note that we do not provide any claim on the impact of the full dataset size $N$ on the magnitude of the Shapley value. We have changed the description of the caption in Fig. 1 to make it clearer. Thanks for pointing this out.
>
> > should there be a $\forall i\in N$ there? The conclusion involves bounding all $\psi_i$ for all $i \in N$
>
> Yes, you are right that we are bounding this for all $i$. We have added this in our updated PDF. Thank you for pointing out.
>
>
> > Can you discuss what technical challenges there were for obtaining the high-probability guarantees and bounding query complexity, given prior work like Reda et al for the linear setting and past work on Gaussian process bandits?
>
>
> The proof of our theoretical results in Theorem 4.1&4.2 (i.e., high probability guarantee) and Theorem 4.3 (i.e., query complexity) requires additional tricks (which we will describe correspondingly) compared to the linear setting (Reda et al., 2021) and the Gaussian process bandits (Chowdhury & Gopalan, 2017): 1) Our paper aims to identify the top-$m$ arms while the Gaussian bandit paper (Chowdhury & Gopalan, 2017) aims to identify the best arm and hence different acquisition functions; 2) The linear top-$m$ arm bandit (Reda et al., 2021) does not consider the non-linear case and did not provide results on query complexity.
>
> For Theorem 4.1&4.2, the work of Chowdhury & Gopalan, 2017 derives the confidence bound for the difference between GP posterior mean and the ground truth reward function while in our paper, we derive the confidence bound for the difference of the predicted gap $\mu_t(z_i) - \mu_t(z_j)$ and the true gap $\psi_i-\psi_j$ due to the difference in acquisition function. On the other hand, the focus of Chowdhury & Gopalan, 2017 is on the analysis of the regret of the algorithm while our Theorem 4.1&4.2 analyzes the correctness and bounds on the stopping iteration of our algorithm in identifying the $(\epsilon, \delta)$-approximation of the top-$m$ data values which requires additional tricks to leverage the result from Reda et al., 2021.
>
> For Theorem 4.3, Reda et al., 2021 did not derive any query complexity for top-$m$ linear bandits while we leverage the complexity of the growth of information gains from Chowdhury & Gopalan, 2017 to derive a query complexity for our algorithm and hence was not seen in previous work.
>
>
> ---
>
> Thank you again for your time and your careful feedback. We hope our clarifications and additional results could improve your opinion of our work.

---

> ### Author Response · Authors · 2024-11-25
> **We would like to know if you have any further questions that require additional clarifications.**
>
> Dear Reviewer 7tcQ,
>
> Thank you again for your time in reviewing our paper and for asking many valuable questions.
>
> Please let us know if our replies have addressed your concerns. We would be happy to address any further questions you might have within the discussion period.
>
> Best regards,
>
> Authors

---

> ### Author Response · Authors · 2024-11-30
> **Thank you for taking the time to review our paper!**
>
> Thank you for taking the time to review our paper!
>
> As the discussion period is concluding in 2 days, we hope to hear from you whether our rebuttal has sufficiently addressed your questions and concerns. We have provided both theoretical and empirical justification for the use of the Shapley value in data selection, noise data detection, and data debugging. We have also provided additional experiments and discussion on the diminishing return and additional discussion on the challenges in the proof of our theoretical results. We hope that our additional results and discussion helped to improve the quality of the paper.
>
> We are more than happy to answer any further questions during the remaining discussion period.
>
> Best regards,
>
> Authors

---

### Author Response · Authors · 2024-12-04
**Summary of the discussion during the rebuttal period**

We sincerely appreciate the efforts of all our dedicated reviewers. The constructive feedback in the reviews significantly enhanced the quality of our paper. We are very grateful!

We are happy to know that most reviewers have acknowledged that their concerns were addressed during our discussion period. Please refer to the individual responses for detailed explanations, and we provide a summary of the discussion here:


- On the design choice of using GP [**Reviewer gPqR and Reviewer kM2u**]: We have provided both empirical and theoretical justification of why GP is a good choice in modeling data values, we have also discussed the advantage of GP over neural networks.

- On the improvement of memory and computational efficiency of GP [**Reviewer gPqR and Reviewer kM2u**]: We have discussed our implementation of GP to improve the computational efficiency and provided reference to these existing works on GP that can be directly used to improve the memory and computational efficiency of GP in our algorithm.

- On the diminishing return [**Reviewer 7tcQ and Reviewer kM2u**]: We have provided additional empirical verification of diminishing return for data points with top-$m$ data values and discussed its relationship with our theoretical results in our response for Reviewer 7tcQ. We have discussed the factors that affect the rate of diminishing of the return in the response for Reviewer kM2u.

- On the use of Shapley value for data selection and other applications [**Reviewer 7tcQ**]: We have provided justification (both theoretical and empirical verification from existing works) on the ubiquity and effectiveness of using Shapley value for data selection, noise data detection, and data debugging.

- On the technical challenges of the proof of our theoretical results [**Reviewer 7tcQ**]: We have pointed out the significant challenges that can not be directly solved from the proof of existing works and described our ways to tackle these challenges in our proof.

- On the assumption of sub-Gaussian [**Reviewer gPqR**]: We have provided additional discussions on why the sub-Gaussian holds for common utility functions used in data valuation even if the utility function is not bounded.

- On the choice of hyperparameters [**Reviewer gPqR**]: We have provided additional explanations on why all theoretical results hold with a single set of hyperparameters and how to choose $\lambda$ to achieve theoretical guarantee and empirical results.

- On the comparison of query complexity with existing baselines [**Reviewer PSf4**]: We have provided the complexity of approximating Shapley value (instead of finding top-$m$) directly using Monte-Carlo-based methods which serves as a peek at the theoretical improvement of query complexity of our method over existing baselines. We have also provided a discussion on the challenges of comparing the query complexity directly with existing baselines.


We are happy to see that most concerns mentioned above have been addressed which the reviewers have acknowledged in their response during the discussion period. We have updated our PDF to incorporate these useful suggestions. Again, we would like to extend our thanks to the reviewers for their constructive feedback and valuable insights.

Best wishes,

Authors

---

### Meta-Review · Area_Chair_mT1f · 2024-12-21

**Metareview:**

This paper introduces the GPGapE algorithm, combining Gaussian Processes with top-m best arm identification. Theoretical analysis covers sample complexity and PAC guarantees. A simplified version of GPGapE is also explored, reducing computational complexity while achieving even better performance in numerical experiments.

This paper addresses a relevant problem where the integration of Gaussian Processes (GP) with top-m best arm identification is a natural and effective approach. Both the theoretical analysis and experimental results are solid, providing a strong foundation for the proposed method. However, there is a minor concern regarding the setting of the GP parameter, which requires knowledge of an upper bound on the sample complexity to achieve theoretical guarantees. Introducing the idea of using diminishing returns to simplify the GPGapE algorithm is also novel.

Considering the above novelties, I recommend acceptance of the paper; please incorporate all the reviewer's feedback in the final version.

**Additional Comments On Reviewer Discussion:**

Summary of the discussion during the rebuttal period.

---

### Decision · Program_Chairs · 2025-01-22

Accept (Poster)